# Brain tumor classification in VIT-B/16 based on relative position encoding and residual MLP

**Shuang Hong** ⬚ *, **Jin Wu, Lei Zhu, Weijie Chen**

School of Information Science and Engineering, Wuhan University of Science and Technology, Wuhan, Hubei, China

* hongshuang1206@wust.edu.cn

## Abstract

Brain tumors pose a significant threat to health, and their early detection and classification are crucial. Currently, the diagnosis heavily relies on pathologists conducting time-consuming morphological examinations of brain images, leading to subjective outcomes and potential misdiagnoses. In response to these challenges, this study proposes an improved Vision Transformer-based algorithm for human brain tumor classification. To overcome the limitations of small existing datasets, Homomorphic Filtering, Channels Contrast Limited Adaptive Histogram Equalization, and Unsharp Masking techniques are applied to enrich dataset images, enhancing information and improving model generalization. Addressing the limitation of the Vision Transformer's self-attention structure in capturing input token sequences, a novel relative position encoding method is employed to enhance the overall predictive capabilities of the model. Furthermore, the introduction of residual structures in the Multi-Layer Perceptron tackles convergence degradation during training, leading to faster convergence and enhanced algorithm accuracy. Finally, this study comprehensively analyzes the network model's performance on validation sets in terms of accuracy, precision, and recall. Experimental results demonstrate that the proposed model achieves a classification accuracy of 91.36% on an augmented open-source brain tumor dataset, surpassing the original VIT-B/16 accuracy by 5.54%. This validates the effectiveness of the proposed approach in brain tumor classification, offering potential reference for clinical diagnoses by medical practitioners.

**Data Availability Statement:** All Datasets files are available from the Figshare and Kaggle site database. https://figshare.com/articles/dataset/brain_tumor_dataset/1512427 https://www.kaggle.com/sartajbhuvaji/brain_tumor_classification-mri/

## 1. Introduction

Cancer remains a leading cause of non-natural deaths in human society. According to statistics released by the National Cancer Center, in 2016, out of the 4.064 million cancer patients in China [1], approximately 10.9% were diagnosed with brain tumors. As the most intricate organ in the human body, the brain is susceptible to the generation of brain tumors due to the abnormal growth and functioning of cells within the brain tissue [2]. To determine the treatment plan for patients, early screening and diagnosis of brain tumors primarily rely on

metadata https://www.kaggle.com/datasets/ahmedhamada0/brain_tumor_detection/metadata.

**Funding:** The author(s) received no specific funding for this work.

**Competing interests:** The authors have declared that no competing interests exist.

modern medical imaging technologies. Physicians utilize medical imaging techniques such as X-rays, ultrasounds, CT scans, and MRI scans to identify the location and extent of the lesions within the patient's brain [3]. Indeed, Brain tumors encompass a wide range of tumor types, each with distinct characteristics and behavior, this requires a high level of expertise from physicians [4], extensive clinical experience, and a strong foundation of prior understanding. However, healthcare professionals generally face high work intensity and heavy workloads, which can lead to the possibility of missed diagnoses and misdiagnoses during the screening and diagnosis of brain tumor images [5].

The integration of artificial intelligence technology has led to the widespread adoption of computer-aided diagnosis and treatment techniques, significantly mitigating the occurrence of missed diagnoses and misinterpretations during tumor image screening and diagnosis. Currently, brain tumor classification tasks primarily rely on Convolutional Neural Network (CNN), and there is a limited amount of brain tumor datasets with relatively small-scale open-source datasets. The emergence of Vision Transformers (VIT) [6] has challenged the necessity of CNN for classification tasks. Applying the Transformer module [7] directly to segmented small image sequences, especially after pre-training on a comprehensive image dataset, has demonstrated remarkable outcomes. Vision Transformers excel with limited computational resources, attaining substantial classification accuracy even with modest datasets.

Yet, Vision Transformers face challenges such as the inability of their core self-attention mechanism to capture input token order, limiting effectiveness in structured data modeling. Additionally, the MLP structure within Vision Transformers for classification encounters information degradation, leading to the loss of significant image details and premature performance saturation. To address the limitations of the aforementioned methods, this paper proposes a brain tumor classification approach based on relative position encoding and residual MLP structure. The main contributions of this work are as follows:

1 Enhanced Dataset Utilization with Image Enhancement Techniques: We enhance the brain tumor dataset using advanced image enhancement techniques, including Homomorphic Filtering (HF) [8], All Channels Contrast Limited Adaptive Histogram Equalization (CLAHE) [9], and Unsharp Masking (UM) [10]. These methods are chosen to enhance image details, mitigate noise levels, and improve contrast in brain tumor images [11]. The resulting dataset not only advances the quality of input data but also amplifies the generalization capability of the network. Enabling the Vision Transformer model to leverage data for more accurate classification.

2 Incorporating Relative Position Encoding for MRI Images of Brain Tumors Understanding: A novel relative position encoding method is introduced during network training, encoding the relative distances between input tokens. This mechanism enables our model to discern and comprehend pairwise relationships between tokens, enhancing its ability to understand the spatial context of brain tumor images.

3 Empowering Vision Transformers through Residual MLP Architecture: The incorporation of this architecture elevates the network's capacity to model complex features within brain tumor images, thereby augmenting the model's classification accuracy. Furthermore, by introducing adaptive average pooling in the fully connected layer of the MLP head, we achieve heightened classification accuracy without inducing a substantial increase in computational complexity [12].

## 2. Related work

In the field of medical-assisted diagnosis, convolutional neural networks have been widely used as deep learning models and have achieved significant results [13]. In 2019, COVID-19 as a pandemic disease has affected millions of human lives and caused a massive burden on healthcare centers. Khan et al [14]. proposed a quick, accurate, and low-cost computer-based tool that's two new deep learning frameworks: Deep Hybrid Learning (DHL) and Deep Boosted Hybrid Learning (DBHL) for timely detection and treatment of COVID-19 patients. After a period of research, Khan et al [15]. also presented a new deep CNN-based framework by novel channel-boosted CNNs to detect and analyze COVID-19 from lung CT images, which can capture useful dynamic features of the infected regions, discriminating the COVID-19 infected region from the healthy ones. The two innovative researches by Khan et al. have provided robust support to the healthcare system at that time, enabling rapid and accurate identification of COVID-19 infection in patients through pulmonary CT imaging. And in the research related to lymphocytes, Rauf et al [16]. proposed a lymphocyte analysis framework based on a deep convolutional neural network (DC-Lym-AF) to analyze lymphocytes in immunohistochemistry images. This framework has the potential to be turned into a medical diagnostic tool to investigate various histopathological problems. Amidst these compelling strides in medical research, malaria cast a significant public health challenge, affecting millions worldwide each year. Khan and Saddam Hussain [17] responded to this pressing concern with the development of an innovative Deep Boosted and Ensemble Learning (DBEL) framework tailored for the screening of malaria parasite images. The essence of their approach involved the strategic amalgamation of new Boosted-BR-STM CNNs and ensemble machine learning classifiers. This fusion of methodologies heralded a novel approach to address malaria's potentially fatal impact on red blood cells.

In brain tumor research, Rehman et al [18]. proposed the use of 3D CNN for brain tumor detection. They employed a feedforward neural network to select the optimal features for classification. While their approach achieved commendable accuracy, it's important to acknowledge the time-consuming nature of the detection process and the complexity associated with their classification procedure. These limitations underscore the need for more efficient methods that strike a balance between accuracy and computational efficiency. Zhao et al [19]. introduced a feature fusion layer into the original U-Net network for brain glioma classification. This approach aimed to create an end-to-end classification system by merging shallow and deep features. However, it tended to overexpress redundant features and did not fully utilize the images' global and local salient features, thereby leaving room for further improvement in classification accuracy. Indeed, brain tumor classification is challenging because of its complex structure, texture, size, location, and appearance. Zahoor, M. M. et al [20]. developed a novel deep residual and regional-based Res-BRNet Convolutional Neural Network for effective brain tumor Magnetic Resonance Imaging (MRI) classification, they have achieved excellent performance in image classification tasks. However, there is still room for improvement in both the computational complexity and efficiency of this model.

In light of the limitations associated with the aforementioned approach, ViT divides the input image into a set of fixed-size image patches or blocks, which are then transformed into sequential data. It operates on the self-attention mechanism, allowing it to capture long-range dependencies and relationships between image patches [21]. This global contextual understanding is especially valuable for brain tumor analysis, where capturing intricate patterns and relationships across different regions of the image is crucial. Traditional CNN, on the other hand, relies on local receptive fields, which might struggle to capture global contextual information effectively. And compared to traditional CNN, ViT can achieve comparable or even

better performance with fewer parameters. This can lead to improved computational efficiency during training and inference, which is important for medical applications where computational resources might be limited. Introducing the ideas of Transformers provides a new paradigm for image processing, allowing models to learn features and relationships directly from raw pixel-level data without relying on handcrafted feature extractors. This innovative model structure opens up new avenues and insights for further exploration of Transformer applications in visual tasks. For instance, in the field of medical image segmentation, Khan et al [22]. propose a novel medical image segmentation technique based on Vision Transformer. This method effectively models dependencies between distant structures through a multi-scale attention mechanism, providing a successful solution to the challenges of segmenting complex, interconnected structures.

## 3. Materials and methods

In this study, the brain tumor images are preprocessed using the HF, CLAHE, and Unsharp Masking image enhancement methods. The relative position encoding method is applied to generate weights for the positional relationships between input tokens. The performance of VIT-B/16 is optimized by incorporating a residual structure in the MLP. The overall model architecture is illustrated in Fig 1. From the model architecture diagram, it can be observed that the model is based on the VIT-B/16 network and is divided into three parts: the Patches

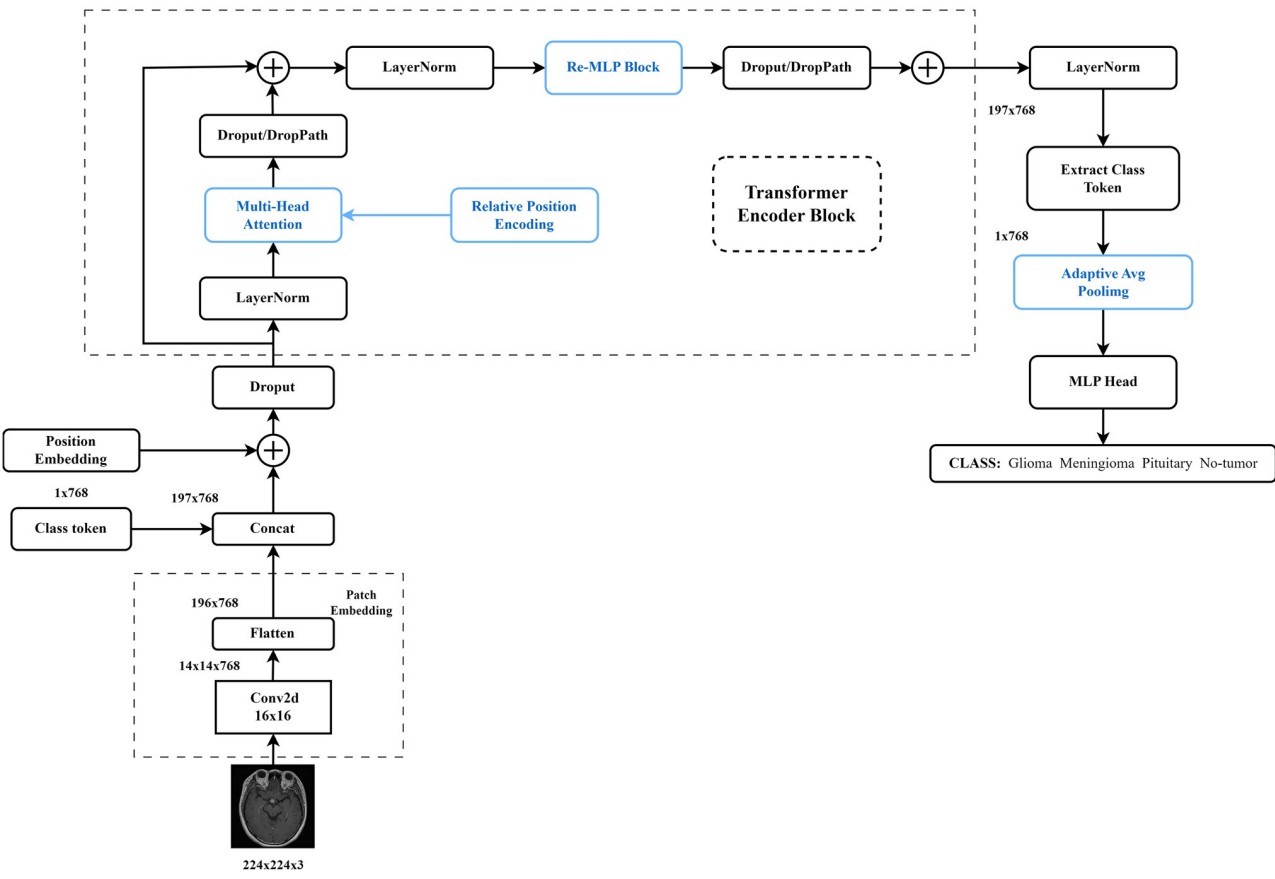

**Fig 1. Improved Vision Transformer network structure.**

Embedding layer, the Transformer Encoder layer with repeated stacked Encoder Blocks, and the MLP Block. Before being input to the Transformer Encoder, these tokens need to be inserted with a trainable vector called the Class Token, which is specifically designed for classification purposes. To address the inherent limitation of self-attention in capturing the order of input tokens, a new two-dimensional image encoding method called Image-Relative Position Encoding (IRPE) is employed for the multi-head self-attention mechanism of the Transformer [23]. The interaction computation between the query, value, and key in the self-attention module is performed by incorporating learnable parameters.

In IRPE, the clip function is introduced to map relative positions to encodings, reducing computational costs and the number of parameters. Later, Dropout/Drop Path is added to accelerate computation speed and improve the model's generalization ability [24]. Brain tumors of different types and stages exhibit varying degrees of lesion severity, and most of them do not exhibit significant differences from a normal brain on MRI. This makes it easy for the model to overlook crucial features, leading to the quick disappearance of gradients and a decrease in recognition accuracy. To address this issue, an improved residual MLP is introduced to enhance the transmission of feature information within the network and improve its performance. The residual structure of the MLP increases the depth of the network and thus its complexity, to address this issue, an adaptive average pooling layer is added to the MLP Head, which helps reduce the number of parameters and computational overhead to some extent. Finally, the classification results are obtained through a fully connected layer with a tanh activation function. Our source code will be made publicly available at: https://github.com/zhulei2016/RST-saliency/upon_acceptance.

### 3.1 Dataset

The dataset used in this paper consists of a few open-source CE-MRI datasets, which are a combination of the Figshareand Kaggle site [25]. The original dataset used in this study comprises a total of 7023 MRI images of brain tumors. Specifically, it includes 1621 images of glioma slices, 1645 images of meningioma slices, 1757 images of pituitary tumor slices, and 2000 images of tumor-free slices. Each tumor type is represented in three different orientations: axial, sagittal, and coronal, as shown in Fig 2.

To address the limited number of samples in the dataset, this paper employs image augmentation to expand it, thereby mitigating overfitting and improving the accuracy and generalization of the network. Three image enhancement methods, namely HF, CLAHE, and Unsharp Masking are used to generate additional images. This augmentation process results in a total of 14,046 images. All images are then standardized by resizing them to a uniform size of 224x224 pixels and normalizing their pixel values. To evaluate the performance of the model, a random split is performed, with 80% of the images used as the training set and the remaining 20% as the test set as shown in Table 1.

### 3.2 Image enhancement

Image enhancement techniques start with the initial MRI images and aim to enrich the image information, enhance image quality, and meet the input requirements of subsequent models while improving model accuracy [26]. We employed the techniques of Homomorphic Filtering (HF), Contrast Limited Adaptive Histogram Equalization (CLAHE), and Unsharp Masking (UM) in this paper. In brain tumor MRI images, noise and significant variations in brightness often exist, resulting in poor contrast and blurred details. Homomorphic filtering, combined with image space and frequency domain features, enhances the details in the darker regions of the image and improves overall resolution. Histogram equalization is a commonly

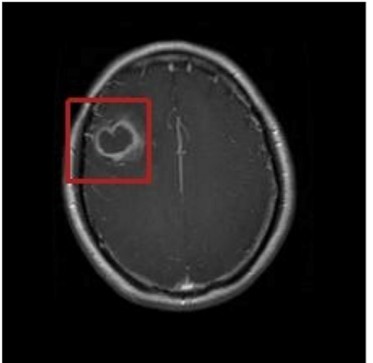 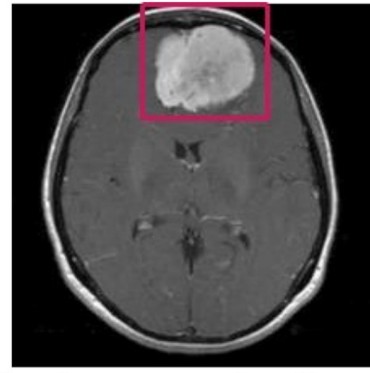 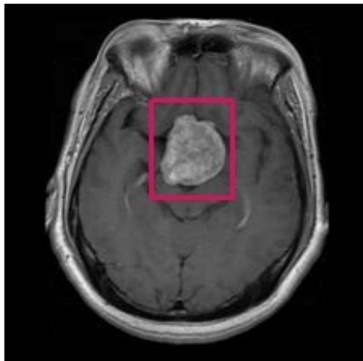

**(a)** The images in the transverse direction correspond to glioma, meningioma, and pituitary tumors

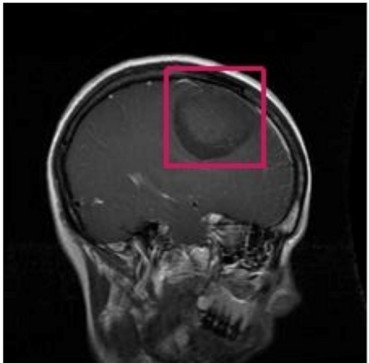 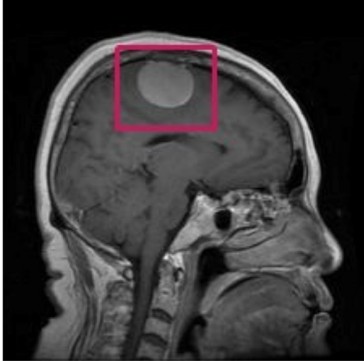 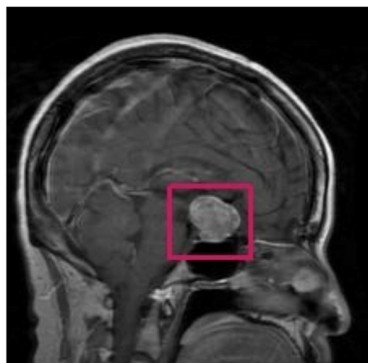

**(b)** The images in the sagittal direction correspond to glioma, meningioma, and pituitary tumors

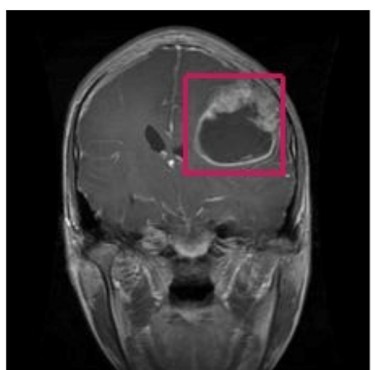 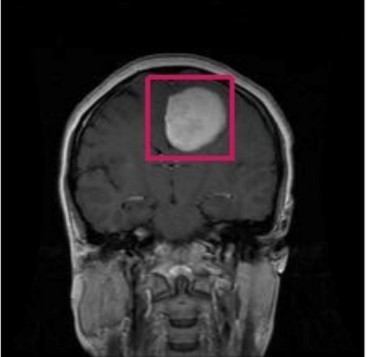 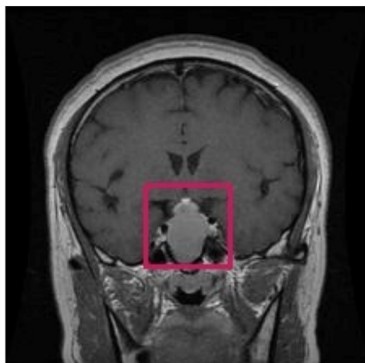

**(c)** The images in the coronal direction correspond to glioma, meningioma, and pituitary tumors

**Fig 2. The examples of images in different orientations for different types of brain tumors.**

**Table 1. Number of samples for each classification.**

| Types | Sample numbers | Training set | Test set |
|---|---|---|---|
| Glioma | 3242 | 2594 | 648 |
| Meningioma | 3290 | 2632 | 658 |
| Pituitary Tumor | 3514 | 2811 | 703 |
| No Tumor | 4000 | 3200 | 800 |

used image enhancement technique. However, traditional histogram equalization methods perform equalization over the entire brightness range of the image, which can result in increased noise and unwanted details, additionally, it may also disrupt the local information of the image. CLAHE improves the overall contrast of an image without negatively impacting the image details. It addresses the limitations of traditional histogram equalization by applying adaptive equalization locally. This approach prevents the amplification of noise while enhancing the contrast. Lastly, the Unsharp Masking technique is applied to enhance the edges and details of MRI images, resulting in sharper image contours.

**3.2.1 Homomorphic filtering.** The homomorphic filtering technique applies the illumination-reflection model to transform each pixel value $f(x, y)$. Each pixel value is equal to the product of the illumination component and the reflection component in this model. The formula is as follows:

$$f(x, y) = r(x, y) \times i(x, y) \tag{1}$$

The reflection component : $0 < r(x, y) < \infty$

The illumination component : $0 < i(x, y) < 1$

The pixel value $f(x, y)$ is subjected to logarithm and Fourier transform, resulting in $F(x, y)$. Then, a Gaussian high-pass filter $H(u, v)$ is applied to enhance the contrast of $F(x, y)$.

$$lnf(x, y) = lnr(x, y) + lni(x, y) \tag{2}$$

ln $r(x, y)$is subjected to Fourier transform, resulting in $R(x, y)$, ln $i(x, y)$is also subjected to Fourier transform, resulting in $I(u, v)$.

$$F(u, v) = R(u, v) + I(u, v) \tag{3}$$

$$H(u, v) \times F(u, v) = H(u, v) \times R(u, v) + H(u, v) \times I(u, v) \tag{4}$$

$$H(u, v) = (\gamma_H - \gamma_L) \times \left\{ 1 - \exp\left[ -c\frac{D^2(u, v)}{D_0^2} \right] \right\} + \gamma_L \tag{5}$$

Where the high-frequency component $\gamma_H = 1.5$, the low-frequency component $\gamma_L = 0.5$, the constant $c = 1$, the cutoff frequency $D_0 = 40$, the distance from the point$(u, v)$to the filter center$(u_0, v_0)$is denoted as $D(u, v)$.

As shown in Fig 3, homomorphic filtering enhances contrast and highlights image details, resulting in a significant improvement in overall resolution. After applying homomorphic filtering to brain tumor MRI images, CLAHE is performed.

**3.2.2 Contrast limited adaptive histogram equalization.** Traditional histogram equalization performs equalization of the overall image brightness [27]. However, the uneven distribution of brightness in brain tumor MRI images can result in the loss of local image information. CLAHE reduces the enhancement magnitude of local contrast by limiting the height of the local histogram to control the amplification of image noise and prevent excessive enhancement of local contrast, which can lead to the loss of details.

CLAHE divides the image to be processed into multiple sub-blocks, where each sub-block contains the number of pixels denoted as S. The histogram of each sub-block is represented by $h(r_k)$, and it is clipped using the corresponding clip threshold value applied to $h(r_k)$, the clipped pixels are redistributed uniformly across the histogram grayscale, and this process is repeated

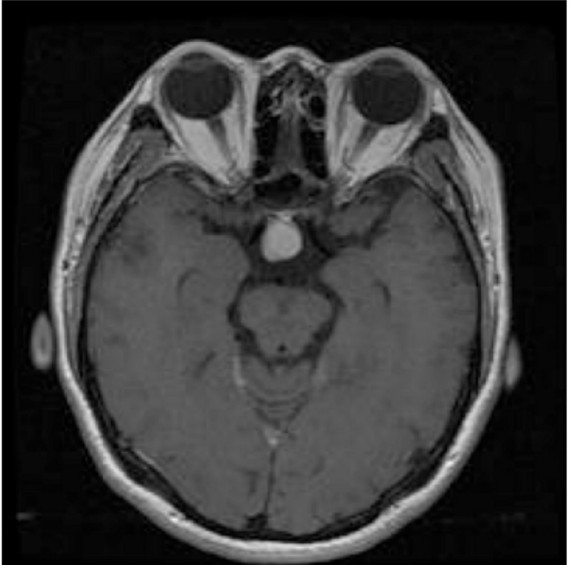

**(a) Original image**  **(b) HF**

**Fig 3. Original image and homomorphic filtered enhanced image.**

until all pixels have been assigned. After the redistribution, the grayscale histogram of each sub-region undergoes histogram equalization. Finally, the grayscale value of each pixel in the output image is calculated based on the processed center pixel of the corresponding sub-region.

$$\text{Clip threshold} : CL = n \times \frac{s}{L}$$

$$\text{Total number of pixels exceeding the CL} : E = \sum_{k=0}^{L-1}\left(max\left(h(r_k)\sum CL, 0\right)\right)$$

$$\text{The histogram after the redistribution} : h'(r_k) = \begin{cases} CL & h(r_k) > CL - B \\ h(r_k) & h(r_k) \leq CL - B \end{cases}$$

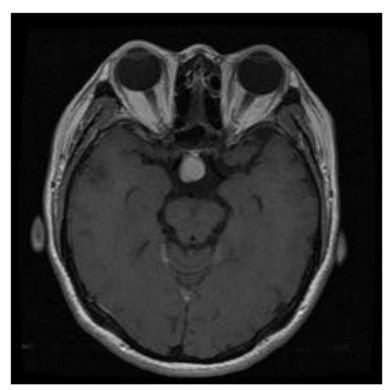 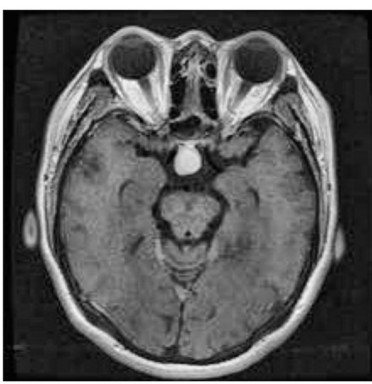 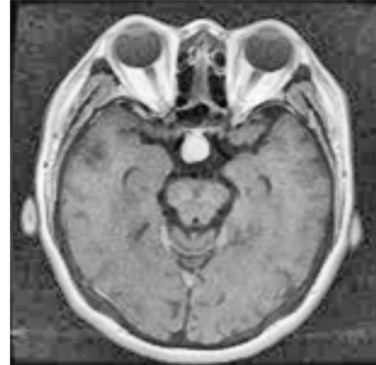

**(a) Original image**  **(b) CLAHE**  **(c) HF+CLAHE**

**Fig 4. Original image, the CLAHE-enhanced image, and the image enhanced by combining HF and CLAHE.**

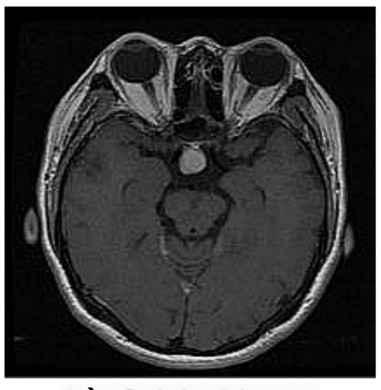 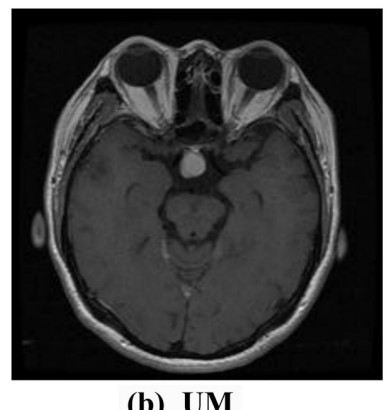 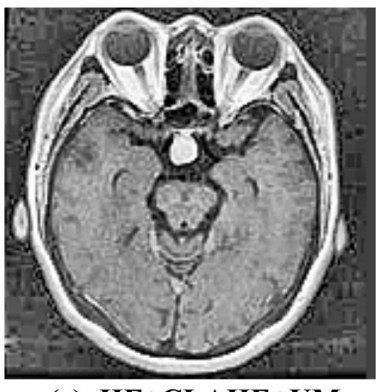

**(a) Original image**  **(b) UM**  **(c) HF+CLAHE+UM**

**Fig 5. Original image, the image enhanced by Unsharp Masking, and the image enhanced by combining HF, CLAHE, and UM.**

As shown in Fig 4, the image undergoes CLAHE processing, where the parameters for histogram equalization are adjusted based on the characteristics of different parts of the image. This effectively enhances the contrast and details of the image, further improving the quality and interpretability of the deep learning network. It enables the network to better learn and extract image features while reducing the risk of overfitting and the impact of noise.

**3.2.3 Unsharp Masking.** Unsharp Masking is applied to brain tumor MRI images to enhance image features and strengthen image edges, Because there exists essential feature information between human brain tissues and structures, while MRI images often suffer from edge blurring.

Unsharp Masking utilizes a high-pass filter to extract the high-frequency components of the image [28]. These high-frequency components are amplified and then added back to the original image. This process enhances the high-frequency information, such as edges and details, in the original image while preserving the low-frequency information.

As shown in Fig 5, the sharpening process applied to the brain tumor MRI image further enhances the edge and detail features between the tissues. The overall clarity and accuracy of the image are significantly improved. This enhancement facilitates faster convergence and training of the network, thereby improving the model's robustness and generalization capability [29].

The information entropy represents the richness of details contained in an image. The more information the image contains, the higher its information entropy [30]. The evaluation results of information entropy for several methods are shown in Table 2. The method combining HF, CLAHE, and UM shows the highest information entropy, indicating the image enhancement effect is evident.

**Table 2. Information entropy evaluation metric.**

| image | Original image | HF | CLAHE | UM | HF+CLAHE+UM |
|---|---|---|---|---|---|
| Glioma | 5.674 | 6.138 | 4.720 | 5.659 | 7.565 |
| Meningioma | 6.013 | 6.369 | 7.088 | 6.109 | 7.711 |
| Pituitary Tumor | 6.146 | 6.890 | 7.314 | 6.312 | 7.639 |
| No Tumor | 4.372 | 4.439 | 6.380 | 4.609 | 6.788 |

## 3.3 Relative position encoding methods

The Transformer Encoder layer consists primarily of Layer Norm, Multi-Head Attention, Dropout / Drop Path, and MLP Block. Self-attention is the core of the Transformer. As shown in Fig 6, It models the relationships between tokens sequentially to map the query and a set of key-value pairs to an output. Before being inputted into the network, brain tumor MRI images are subjected to block-wise segmentation and flattening. Then, they are mapped to a representation $a = (a_1, a_2 . . .a_n)$. Each element $a_1, a_2. . .a_n$ is individually processed using three-parameter matrices $W^Q$, $W^K$, $W^V$ to compute their corresponding q (query, which is matched with each k), k (key, which is matched with each q), and v (value, which contains extracted information from a). The matching process between q and k calculates their correlation, and the higher the correlation, the larger the weight assigned to the corresponding v.

The weight coefficients $\alpha_{ij}$ are calculated using softmax:

$$\alpha_{ij} = \frac{e^{e_{ij}}}{\sum_{k=1}^{n} e^{e_{ij}}} \tag{6}$$

Where:

$$e_{ij} = \frac{(a_i W^Q)(a_j W^K)^T}{\sqrt{d}} \tag{7}$$

The value of $e_{ij}$ is calculated by scaled dot-product attention, where the dot product may result in large values. This can cause gradients to become very small after applying softmax, so we use sequence length for $\sqrt{d}$ the scaling.

Self-attention computes an output sequence $z = (z_1, z_2, . . .z_n)$, where each output element $z_i$ is computed by a weighted sum of the input elements.

$$z_i = \sum_{j=1}^{n} \alpha_{ij}(a_j W^V) \tag{8}$$

The formula is summarized as follows::

$$Attention(Q, K, V) = softmax\left(\frac{QK^T}{\sqrt{d}}\right)V \tag{9}$$

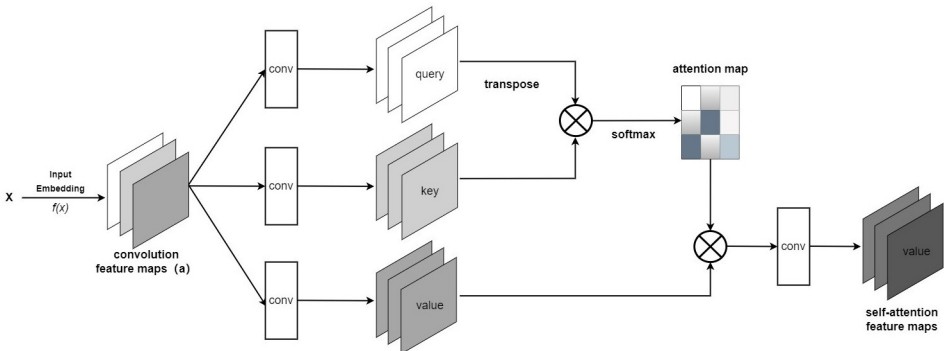

**Fig 6. Self-attention.**

Multi-Head Attention is a crucial component in the Transformer architecture and plays a key role in brain tumor classification. It essentially performs parallel computations of multiple self-attentions, using multiple attention heads, and concatenates their outputs for linear transformation to obtain the desired dimensions. In the processing of brain tumor MRI images, each attention head can capture different aspects of information, providing multiple representation subspaces. This helps in gaining a more comprehensive understanding of brain tumors. For brain tumor MRI images, we utilize randomly initialized Q, K, and V weight matrices in each Attention Head. As shown in Fig 7(b), Q, K, and V undergo linear transformations through the Linear layers before entering the Scale Dot-Product Attention layer (as shown in Fig 7(a). In the Scale Dot-Product Attention layer, Q and K are matrix multiplied using Matmul, followed by dimension scaling in the scale layer. This process allows us to obtain the correlations between Q and each K, which are then used to calculate the softmax-weighted matrix. In the task of brain tumor classification, we can consider these weight matrices as the similarities between image blocks, allowing us to perform a weighted fusion of the image blocks and emphasize features related to brain tumors. During the training phase, we can also utilize a Mask layer to mask out irrelevant sequence information. In brain tumor MRI images, some irrelevant image blocks or noise may exist. Using the Mask layer, we can suppress these interfering factors and enhance the model's focus on the features relevant to brain tumors. Finally, the output is obtained by multiplying the softmax-weighted matrix with V.

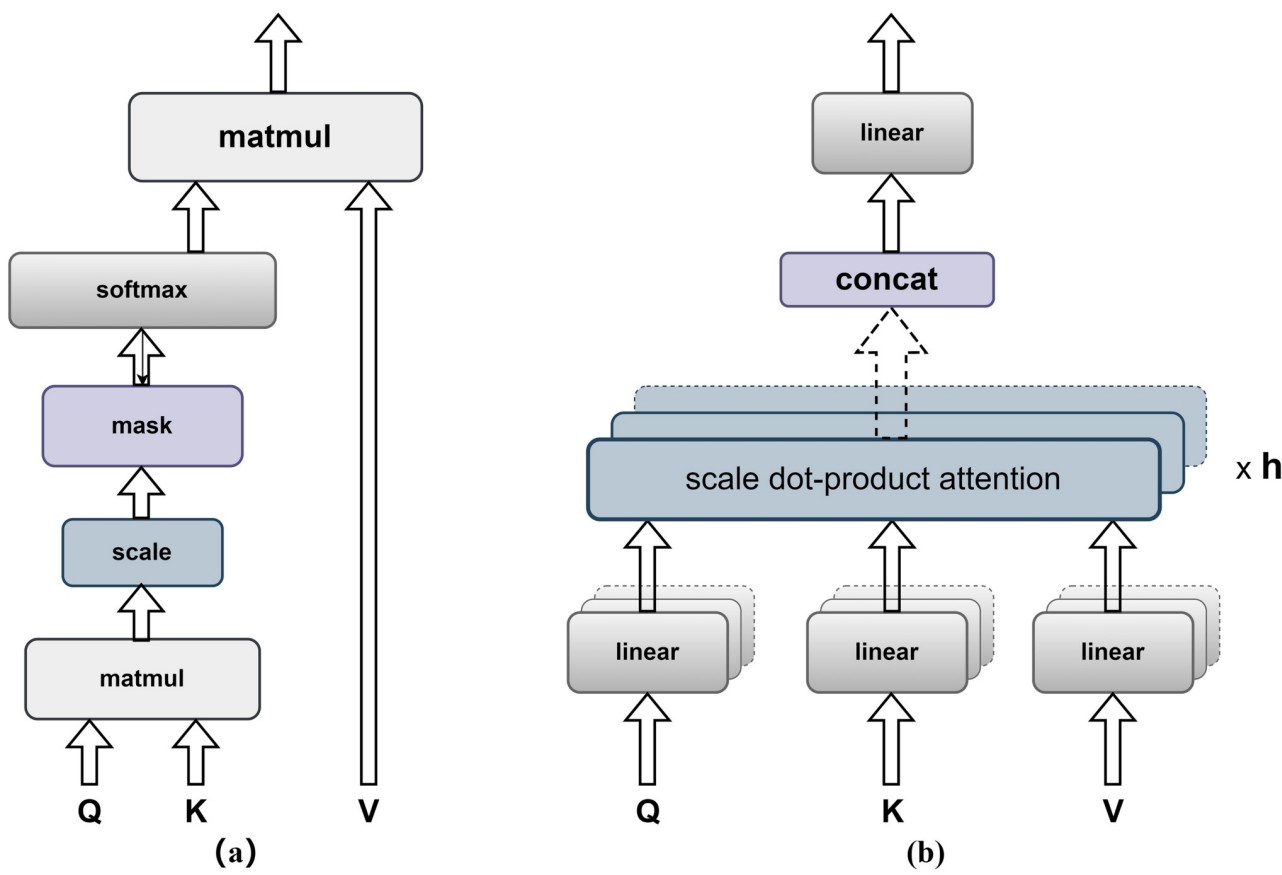

**Fig 7. (a) Scale Dot-Product Attention (b) Multi-Head Attention.**

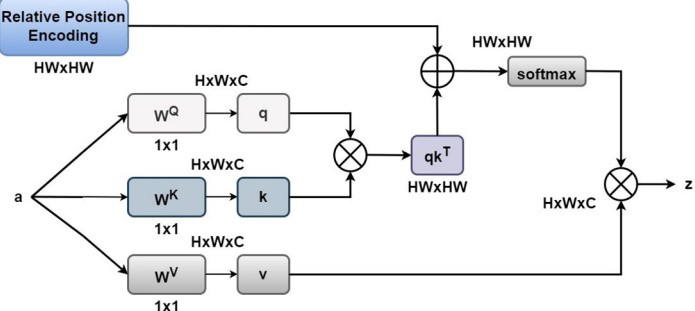

**Fig 8. Relative position encoding.**

In the task of brain tumor classification, brain tumor MRI images have specific spatial structures and positional information. These pieces of information are crucial for accurately capturing the features and dependencies of brain tumors. However, traditional self-attention mechanisms have inherent limitations in capturing the sequential order of input tokens. To address this issue, we introduce the relative positional encoding method to encode the relative distances between input elements. This allows us to learn the positional relationships between input tokens, particularly capturing longer dependencies between tokens. In the processing of brain tumor MRI images, this method encodes the relative positional information between input elements $a_i$ and $a_j$ into vectors $p_{ij}^Q$, $p_{ij}^K$, and $p_{ij}^V$, which are then combined with self-attention. By weighting the dot product results of the query and key with the relative positional information, the consideration of positional relationships is introduced in the attention computation. This approach enables the Transformer model to better understand the dependency relationships between different positions in brain tumor MRI images and more accurately capture important tumor features.

The output $z_i$ element is represented as follows:

$$z_i = \sum_{j=1}^{n} \alpha_{ij}(a_j W^V + p_{ij}^V) \tag{10}$$

Where $\alpha_{ij} = \frac{e^{e_{ij}}}{\sum_{k=1}^{n} e^{e_{ij}}}$ does not change

$$And \ e_{ij} = \frac{\left(a_i W^Q + p_{ij}^Q\right)\left(a_j W^K + p_{ij}^K\right)^T}{\sqrt{d_z}} \tag{11}$$

The approach used in this paper is an independent relative position encoding method, separate from the input embedding layer. In the input embedding layer, the interaction between query, key, and value takes place, while the relative position encoding, which captures the positional relationships between tokens, does not participate in the interaction. Instead, it is added to the dot product result of the query and key before the softmax operation, as shown in Fig 8. In addition, we observed that not all relative positional information between input elements is useful. In brain tumor MRI images, distant positional information is often redundant and can increase the number of model parameters and computational costs. Therefore, we apply the clip function to limit the distance of relative positions, retaining only the relative positional information within a certain range [31]. By doing so, we can more effectively utilize the limited

positional information and improve the performance of the brain tumor classification model. Summing up, we can express it with the following formula:

$$e_{ij} = \frac{\left(a_i W^Q + p_{ij}^Q\right)\left(a_j W^K + p_{ij}^K\right)^T + b_{clip(i-j,k)}}{\sqrt{d_z}} \tag{12}$$

where $b_{clip(i-j, k)}$ is a learnable scalar and is obtained by applying the clip function to the two-dimensional relative positional encoding.

And the clip function:

$$clip(i - j, k) = max(k, min(k, i - j)) \tag{13}$$

The rationale behind introducing relative position encoding is rooted in the recognition of the importance of sequential information in medical imaging. In scenarios like brain tumor classification, the spatial arrangement and order of image patches can carry valuable diagnostic insights. Conventional self-attention mechanisms, while powerful, lack the inherent capability to understand the sequential context. The relative position encoding method addresses this gap by providing an explicit mechanism for the model to comprehend the sequential relationships. Our method involves applying the clip function to two-dimensional relative positional encodings, allowing the model to differentiate between tokens based on their positions and maintain a notion of order. By introducing such positional awareness, the model can better capture the spatial layout of features and thereby improve its capacity to recognize intricate patterns within brain tumor images. In medical imaging classification tasks, the arrangement of features often carries crucial diagnostic information. By equipping the model with the ability to consider positional context, we enable it to harness sequential patterns that could significantly enhance its diagnostic accuracy.

## 3.4 Residual MLP

In Vision Transformer, the MLP (Multi-Layer Perceptron) is used as a sub-module of the self-attention layer to achieve non-linear mapping. It projects the feature vectors of each position in the input to a higher-dimensional space, followed by dimension reduction. Finally, a non-linear activation function like GELU is applied to transform the feature vectors. The MLP can be seen as a fully connected neural network and helps to improve the overall representational capacity of the model [32].

In Vision Transformer, the Multi-Layer Perceptron (MLP) is utilized as a component of the self-attention layer to perform the non-linear mapping. This module projects each position's feature vectors in the input to a higher-dimensional space, followed by dimension reduction. A non-linear activation function, such as the Gaussian Error Linear Unit (GELU), is then applied to the feature vectors. The MLP, resembling a fully connected neural network, enriches the overall representational capacity of the model [32]. A significant enhancement in our approach, depicted in Fig 9, is the introduction of a residual structure into the MLP. Rather than using the original MLP module, we employ a Residual MLP, which utilizes the output from the Multi-Head Self-Attention as input. This modification is not arbitrary but is driven by the need to address the degradation problem typically associated with the original MLP network. In the conventional MLP setup, the input vector first passes through a fully connected layer, which alters the number of input nodes by a factor of four. A GELU activation function [33] and a dropout layer are then applied. The GELU activation function introduces stochastic regularity, enhancing non-linearity in the network while maintaining the integrity of the input data. This process improves the model's generalization ability by preventing overfitting. The

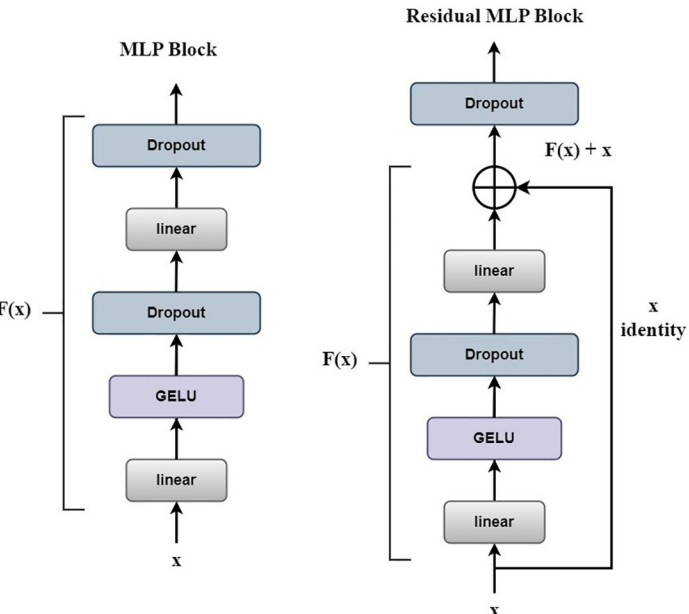

**Fig 9. Original model and improved structure.**

input subsequently passes through another fully connected layer to restore the node number, followed by a dropout layer to produce the final output.

However, with the original MLP, we observed that the network often converged too swiftly, which could lead to suboptimal learning of complex features. This issue is where the residual structure comes into play. In the Residual MLP block, we introduce a shortcut connection that bypasses several layers. This modification transforms the original mapping $F(x)$ into $F(x) + x$, where x is the input vector. The addition of residual structures allows for the learning of more intricate features without significantly increasing the computational burden. The residual structure essentially creates a form of memory in the model, allowing it to learn from previously seen data and thus improving the model's capacity to generalize [34]. Furthermore, the residual structure mitigates the vanishing gradient problem, enabling the model to learn deeper representations without converging prematurely. As a result, we achieve better network recognition accuracy.

## 4. Experiment and analysis

This experiment uses the software python 3.10.8, the deep learning framework pytorch1.13.0 as the backend, the operating system is Windows11, the hardware device CPU is Intel core i7-12700H, GPU is Nvidia RTX3070ti.

### 4.1 The evaluation metrics

The evaluation index accuracy (Accuracy) used in this paper is used to evaluate the performance of our model. The definition of accuracy is as follows:

$$Acc = \frac{N_t}{N} \times 100\% \tag{14}$$

Where $N_t$ represents the number of correctly classified test samples and $N$ represents the total number of test samples.

To comprehensively evaluate the performance of the model, this study also calculates the accuracy for individual classes. The accuracy for a single class is defined as follows:

$$Acc_i = \frac{N_t^{(i)}}{N^{(i)}} \times 100\% \tag{15}$$

Where $N_t^{(i)}$ represents the number of correctly predicted images for a single class by the model, and $N^{(i)}$ represents the total number of images belonging to that class in the test dataset. In addition to the aforementioned two evaluation metrics, we also utilized precision [35], recall [36], F1-score [37], and precision-recall (PR) to assess the effectiveness of the developed model. The definitions are given as Eqs (16) to (18) as shown below:

$$Precision = \frac{TN}{TN + FP} \tag{16}$$

$$Recall = \frac{TP}{TP + FN} \tag{17}$$

$$F1 - score = \frac{2 \times (P \times R)}{P + R} \tag{18}$$

Where TP defined truly positive predictions, TN as truly negative predictions, FP as incorrectly positive predictions, and FN for incorrectly negative predictions.

## 4.2 The training process of the experiments

Based on deep learning theory, a brain tumor image classification task was implemented using the Transformer neural network. The experiment utilized the Adam optimizer with a learning rate of 0.001. A batch size of 8 and 30 epochs was set for the training process. The CE-MRI dataset was employed, and image augmentation techniques were applied to enhance the information of the images, prevent overfitting, and improve the model's generalization ability. Preprocessing steps were also conducted to enhance the model's sensitivity. The overall process of training the brain tumor recognition model is illustrated in Fig 10.

During the experiment, the accuracy and loss rates of the model were recorded and plotted after each epoch. Fig 7 demonstrates the overall trend of the model's accuracy improving over

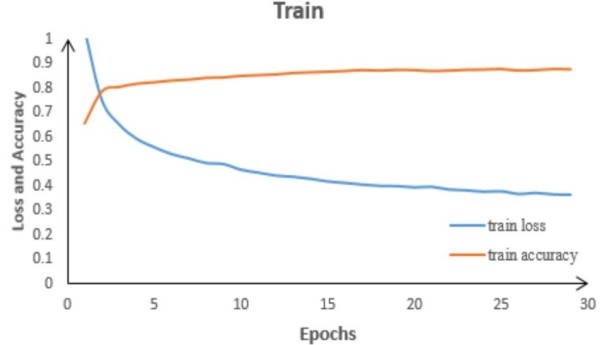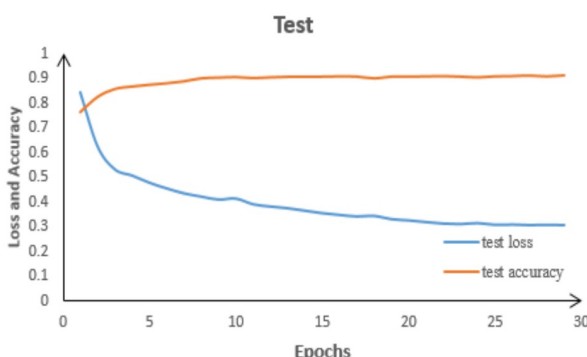

**Fig 10. Training curves of improved VIT-B/16 on training and test sets.**

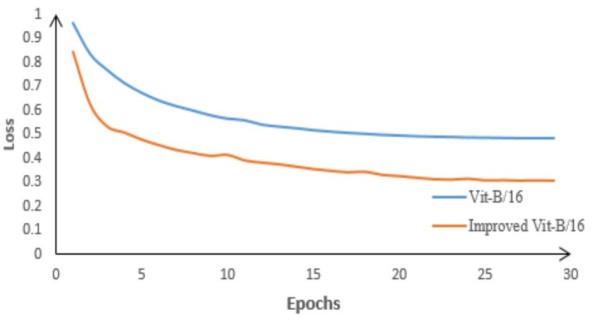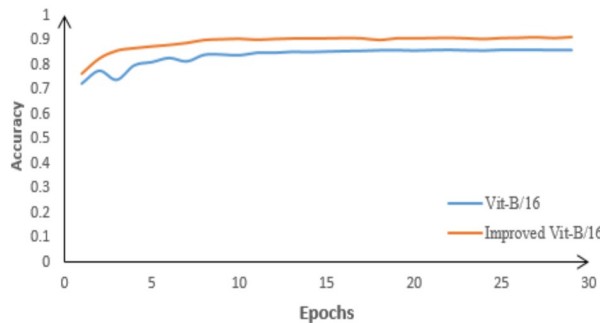

**Fig 11. Loss and accuracy plots of VIT-B/16 before and after improvement.**

iterations, as both the training and testing accuracy rates increase. Simultaneously, the loss rate gradually decreases, indicating that the model is learning and making more accurate predictions as the training progresses. The accuracy of the training set starts to converge around the 10th epoch, and it begins to plateau around the 20th epoch, stabilizing at approximately 0.8765. The loss rate of the training set stabilizes around 0.3651. As for the testing set, the accuracy and loss rates exhibit similar fluctuations to the overall trend observed in the training set. The final accuracy of the testing set is 0.9136, and the loss rate is 0.3064. These values indicate that the model has high overall accuracy and good robustness.

From Fig 11, it is evident that the improved network achieves significantly higher accuracy compared to VIT-B/16. The loss rate of the original network is 0.484, while the improved network has a loss rate of 0.306. This indicates that the improved network exhibits better classification performance for brain tumors. The accuracy of the models before and after the improvement, as well as the accuracy for each individual class, are shown in Tables 3 and 4, respectively.

According to Table 3, it can be observed that the network before the improvement achieves the highest accuracy of 89.38% in classifying normal brain images. However, the accuracy is relatively lower for other types of brain tumor MRI images. This is because the features of the normal brain without tumors are relatively distinct, with no pathological areas in the images. On the other hand, the visual features of other brain tumor images are not easily distinguishable. Some tumor locations may be close in proximity, making it challenging for the network to extract sufficient detailed features. As a result, the classification accuracy for these images is lower. By comparing Tables 3 and 4, we can observe that the improved network achieved an increase in accuracy for each category by 8.64%, 8.25%, 9.67%, and 5.24% respectively. The overall accuracy improved by 5.54%. These results indicate that the improved network demonstrates higher classification accuracy in brain tumor classification tasks.

**Table 3. Classification results of the network before improvement.**

| Types | Samples Numbers | Error Samples | Accuracy |
|---|---|---|---|
| Glioma | 648 | 113 | 82.56% |
| Meningioma | 658 | 117 | 82.21% |
| Pituitary Tumor | 703 | 145 | 79.37% |
| No Tumor | 800 | 85 | 89.38% |
| Total Accuracy | | | 85.82% |

**Table 4. Classification results of improved network.**

| Types | Samples Numbers | Error Samples | Accuracy |
|---|---|---|---|
| Glioma | 648 | 57 | 91.20% |
| Meningioma | 658 | 61 | 90.73% |
| Pituitary Tumor | 703 | 77 | 89.04% |
| No Tumor | 800 | 43 | 94.62% |
| Total Accuracy | | | 91.36% |

Precision and recall (sensitivity) are the primary metrics for evaluating the efficiency of medical assistive diagnosis systems. Effective brain tumor classification demands good classification performance. As depicted in Table 5, our proposed method has been assessed, demonstrating that our model achieved a Precision of 90.6% and a Recall of 90.74%, effectively enhancing the accuracy of the model. The normal distribution plot with the associated 95% confidence interval from Fig 12 provides a comprehensive statistical analysis of your model's accuracy. The shape of the curve reveals the variability in the model's accuracy, while the 95% confidence interval offers an estimate of the true accuracy. Through this visualization, Our model demonstrates relatively stable accuracy across multiple experiments, with the overall performance maintaining a consistent level. Additionally, From Fig 13, through a PR curve analysis, our proposed brain tumor classification model has demonstrated better performance than other classification models on the dataset. The PR curve reveals the model's excellence in terms of precision and recall signifies the achievement of a lower false positive rate, thereby augmenting sensitivity. These demonstrate the robust discriminative capabilities of our model in the intricate landscape of brain tumor classification and our study offers a potential solution for enhancing the accuracy and reliability of brain tumor diagnostic processes.

Fig 14 presents the visual attention maps for the transformer encoder blocks of our model. We randomly selected three types of brain tumor images from the dataset, with each column representing the attention map of a different layer of encoder blocks. We can observe that the model highlights the regions of interest in brain tumor images, with the areas of interest becoming more pronounced as we progress to the final block of the model. This indicates that the model effectively focuses on the locations of lesions in the brain tumor images The visualization results above demonstrate that our model has successfully captured the areas in MRI images that require identification.

### 4.3 Ablation experiments

In this paper, VIT-B/16 is used as the base network in the implementation of brain tumor classification experiments. In this section, the accuracy obtained at each step of the network improvement is analyzed, and ablation experiments are performed for each step of improvement to verify its importance and contribution to the model, and the results of the ablation experiments are shown in Table 6.

The results of the ablation studies demonstrate that by performing image enhancement solely on the dataset, thereby increasing the diversity of the images and providing the network

**Table 5. Performance evaluation of VIT-B/16 and after improvement.**

| Model | Precision | Recall | F1-score | Accuracy |
|---|---|---|---|---|
| VIT-B/16 | 84.25% | 84.50% | 84.37% | 85.82% |
| Ours | 90.60% | 90.74% | 90.67% | 91.36% |

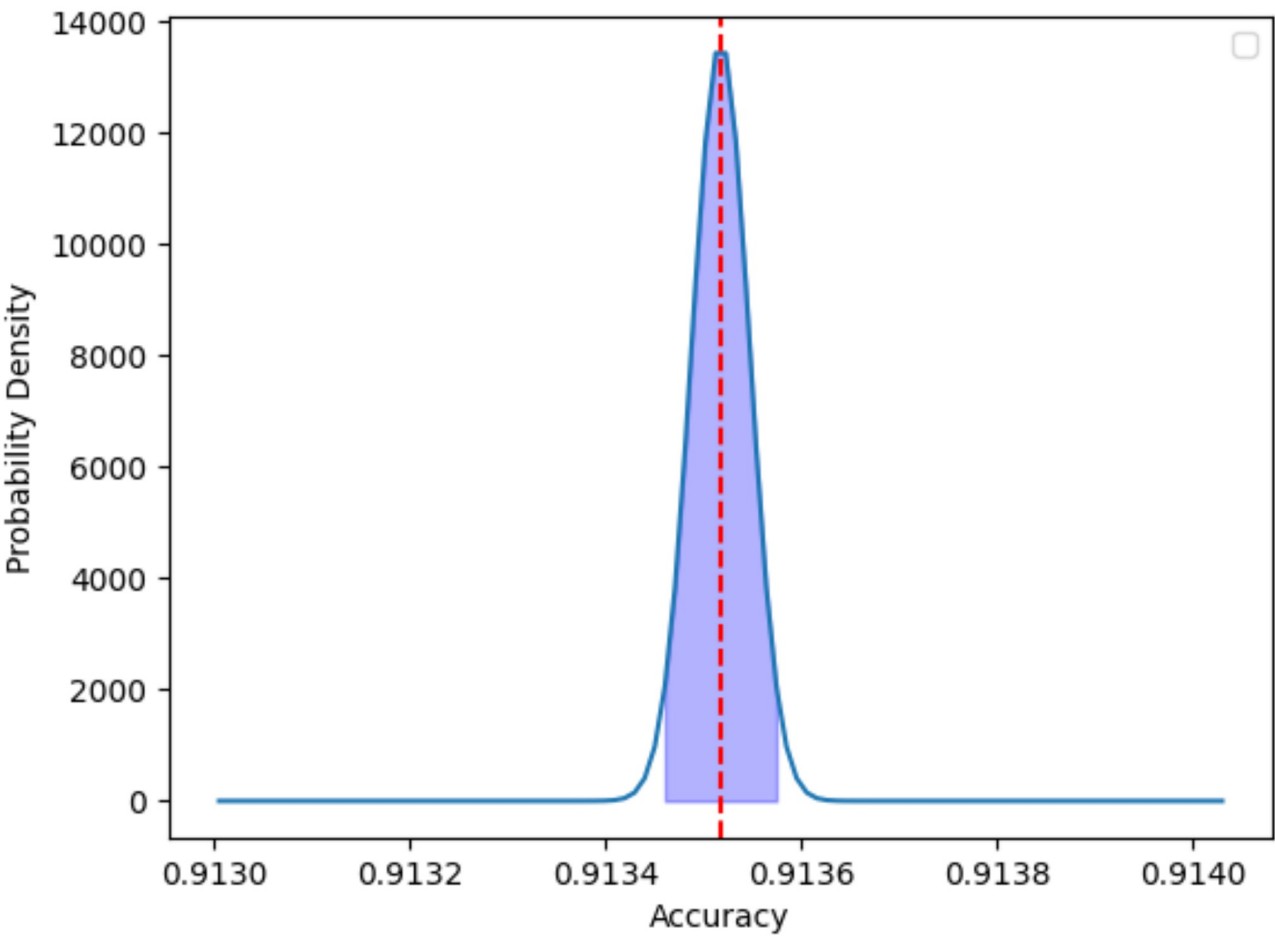

**Fig 12. Normal distribution of accuracy with 95% confidence lnterval.**

with more examples to learn from, the overfitting of the model was mitigated to a certain extent. As a result, the original model's accuracy improved by 0.82%. When only the Residual MLP was added into the network, increasing the depth of the classification network and thereby enhancing the overall model complexity, it was able to better learn complex image features, the accuracy improved to 88.57% as a result. By combining both methods to enrich the image information and enhance the learning capability of the network, the model accuracy improved to 89.51%. Finally, the introduction of a new relative positional encoding method was incorporated to improve recognition accuracy by learning the spatial relationships between different parts of the input images. This enhancement resulted in an overall network accuracy improvement of 5.54% compared to the original network.

### 4.4 Comparison of different models

Table 7 presents a comparison of the performance of the improved network proposed in this study with traditional neural networks, including AlexNet [38], VGGNet [39], GoogLeNet [40], ResNet [41], the lightweight neural network MobileNet [42], and EfficientNet [43] which employs a compound scaling strategy.

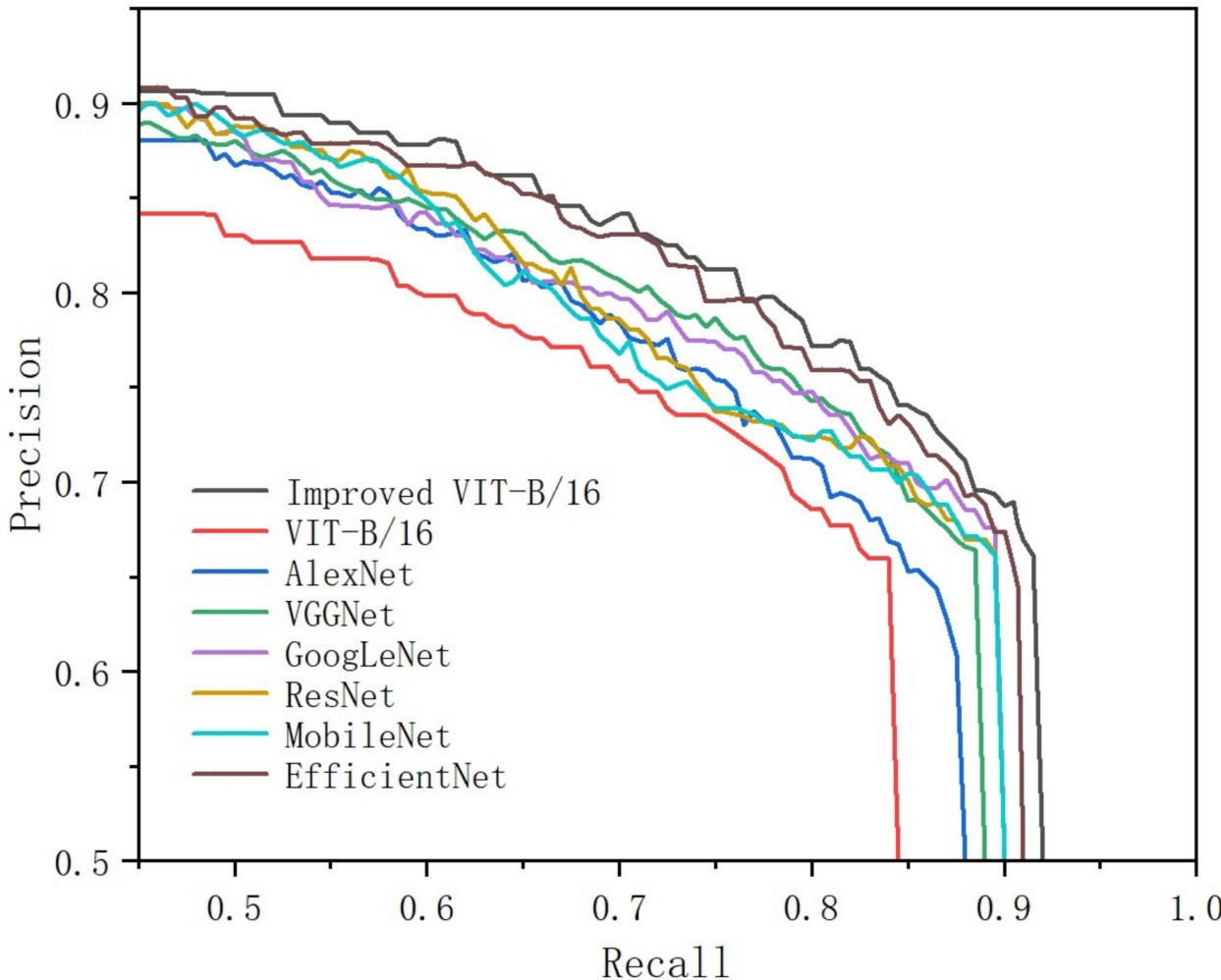

**Fig 13. PR curve based analysis of different classification models.**

From the table, it can be observed that AlexNet demonstrates the effectiveness of deep convolutional neural networks in image classification, achieving an accuracy of 88.02% on this dataset, showcasing the potential of deep convolutional neural networks in the field of image classification. VGGNet, while inheriting from AlexNet, demonstrated that increasing the depth of the neural network while using smaller convolutions can still effectively improve the performance of the network. In this study, VGGNet achieved a model accuracy of 1.71% higher than AlexNet. Like VGGNet, GoogLeNet also utilizes small convolutional layers for dimensionality reduction. In addition, it replaces the fully connected layers with average pooling layers [44]. These modifications contribute to the success of achieving a classification accuracy of over 90% on the brain tumor dataset. VGGNet and GoogLeNet, as traditional neural networks, have achieved performance improvements by stacking convolutional and downsampling layers, but they also suffer from the problem of degradation. In response to this issue, ResNet introduced residual connections to alleviate the degradation problem and enabled the construction of extremely deep network architectures [45]. As a result, ResNet further improved the accuracy to 90.16%. The use of extremely deep network architectures comes

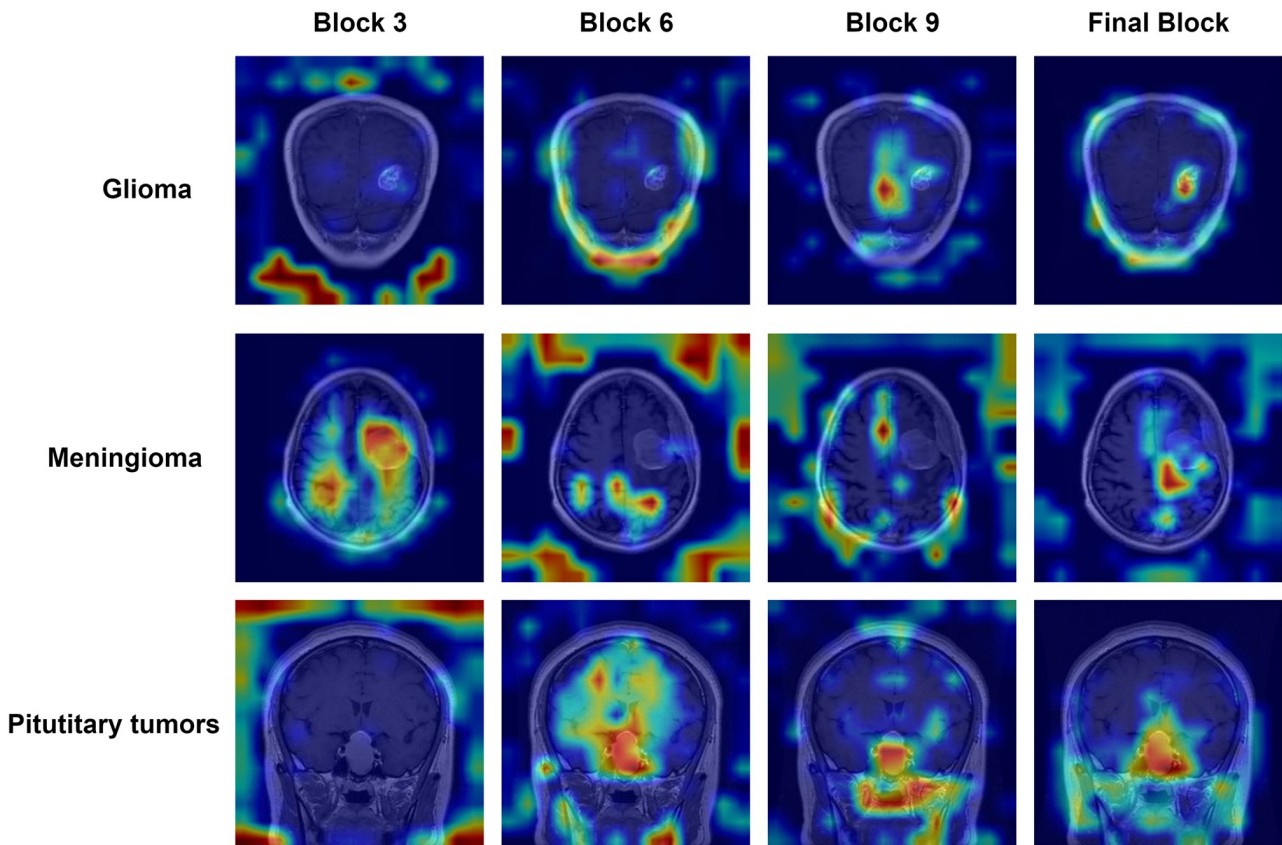

**Fig 14. Visualization results of transformer encoder blocks.**

with the drawback of high memory requirements and computational complexity, which may not be suitable for mobile and embedded applications. The increasing demand for these domains necessitates lightweight network models. MobileNet addresses this requirement by employing the Depthwise Convolution structure, which reduces the computational complexity while maintaining a reasonable level of classification accuracy. The more advanced Efficient-Net architecture incorporates various techniques like depthwise separable convolutions, linear bottleneck structures, and advanced regularization to optimize the trade-off between accuracy and efficiency. This makes them suitable for image classification tasks and also leads to a higher accuracy 90.67% in the classification experiments.

In contrast to traditional convolutional networks, ViT uses self-attention mechanisms to capture global context information from the entire image instead of relying solely on

**Table 6. Results of ablation experiments.**

| Model | Accuracy |
|---|---|
| VIT-B/16 | 85.82% |
| VIT-B/16+ IE | 86.64% |
| VIT-B/16+ Re-MLP | 88.57% |
| VIT-B/16+ IE+ Re-MLP | 89.51% |
| VIT-B/16+ Re-MLP+ Rpe+ IE | 91.36% |

Table 7. Comparison of different classification models.

| Model | Accuracy |
|---|---|
| AlexNet | 88.02% |
| VGGNet | 89.73% |
| GoogLeNet | 90.07% |
| ResNet | 90.16% |
| MobileNet | 90.13% |
| EfficientNet | 90.67% |
| Ours | 91.36% |

convolutional layers [46]., which can be particularly useful for tasks like brain tumor classification where the relationships between different regions of the image are important. ViT generates attention maps that can help in interpreting which parts of the image contribute most to the classification decision. This can be important in medical applications to understand the model's decision-making process. Our improved network achieves an accuracy of 91.36%, which is 3.34% higher than AlexNet, 1.63% higher than VGGNet, 1.2% higher than ResNet, 1.23% higher than MobileNet, and 0.69% higher than advanced EfficientNet. These results demonstrate that our proposed method achieves higher classification accuracy in brain tumor classification.

## 5. Conclusion

Brain tumors, as highly malignant tumors with high incidence and mortality rates, require timely detection and treatment to save patients' lives potentially. Magnetic resonance imaging technology provides excellent visualization of the internal structure and tissue of the brain. With the assistance of classification-assisted diagnostic techniques, doctors can make faster and more accurate judgments regarding the type of tumor, aiding in the efficient diagnosis and treatment of brain tumors. However, the limited size of brain tumor datasets and the scarcity of image sources pose challenges to accurate classification. In this study, image enhancement techniques were employed to enrich the information within the images and increase the number of samples, benefiting network classification accuracy. Traditional classification-assisted diagnostic techniques heavily rely on convolutional neural networks. In this study, ViT-B/16 was employed for brain tumor classification, offering advantages in handling global features, generalization ability, and training speed. The network structure and performance were further optimized by incorporating methods such as relative positional encoding and Residual MLP. The improved network achieved a final accuracy of 91.36%, demonstrating a significant improvement of 5.54% compared to the original ViT-B/16 model and validating the effectiveness of the proposed modifications.

However, it's imperative to acknowledge the inherent limitations and potential pitfalls of deep learning and ViT models, especially when applied to medical applications like brain tumor classification. Datasets are often limited in size and can be biased due to variations in data collection methods, patient demographics, and clinical settings. Such biases can impact model performance and generalization to diverse patient populations. While advancements in generalization have been achieved through techniques like transfer learning, there's still a risk that models might struggle with unseen or rare cases, leading to potential misdiagnoses or inaccuracies in clinical practice. Future research should focus on acquiring more diverse datasets to mitigate biases and enhance model generalization. Advanced data augmentation methods should be explored to artificially increase dataset size and variability, thereby improving

robustness and performance. Integrating MRI data with other medical data types, such as genomic, histopathological, and clinical data, can provide a more comprehensive understanding of brain tumors and enhance classification accuracy. Utilizing transfer learning and domain adaptation techniques can further improve performance on small and diverse datasets. Finally, conducting longitudinal studies to validate model performance over time and across different patient cohorts is essential to ensure reliability and robustness in clinical practice.

## Author Contributions

**Conceptualization:** Shuang Hong.

**Data curation:** Shuang Hong, Jin Wu, Lei Zhu, Weijie Chen.

**Formal analysis:** Lei Zhu, Weijie Chen.

**Investigation:** Shuang Hong.

**Methodology:** Shuang Hong, Jin Wu, Lei Zhu.

**Software:** Shuang Hong, Lei Zhu.

**Supervision:** Jin Wu, Lei Zhu.

**Validation:** Shuang Hong, Jin Wu, Lei Zhu.

**Writing – original draft:** Shuang Hong.

**Writing – review & editing:** Jin Wu, Lei Zhu, Weijie Chen.

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
