## [Decision Letter · Decision Letter 0]

24 Jul 2023

PONE-D-23-19484Brain Tumor Classification in VIT-B/16 based on Relative Position Encoding and Residual MLPPLOS ONE

Dear Dr. Hong,

Thank you for submitting your manuscript to PLOS ONE. After careful consideration, we feel that it has merit but does not fully meet PLOS ONE’s publication criteria as it currently stands. Therefore, we invite you to submit a revised version of the manuscript that addresses the points raised during the review process.

We look forward to receiving your revised manuscript.

Kind regards,

Saddam Hussain Khan

Academic Editor

PLOS ONE

Additional Editor Comments:

The manuscript presents promising results, achieving a 5.54% improvement in classification accuracy over the baseline VIT-B/16 model. However, the evaluation metrics presented are limited to accuracy. It is crucial to include additional performance metrics, such as sensitivity, specificity, precision, and F1-score, to provide a comprehensive assessment of the model's effectiveness. A more in-depth discussion of the results, including the potential impact of the proposed modifications, would enrich the paper and help readers understand the strengths and limitations of the algorithm.

Reviewers' comments:

Reviewer's Responses to Questions

**Comments to the Author**

1. Is the manuscript technically sound, and do the data support the conclusions?

Reviewer #1: Yes

Reviewer #2: Yes

Reviewer #3: No

2. Has the statistical analysis been performed appropriately and rigorously? 

Reviewer #1: Yes

Reviewer #2: No

Reviewer #3: No

3. Have the authors made all data underlying the findings in their manuscript fully available?

Reviewer #1: Yes

Reviewer #2: Yes

Reviewer #3: No

4. Is the manuscript presented in an intelligible fashion and written in standard English?

Reviewer #1: Yes

Reviewer #2: Yes

Reviewer #3: Yes

5. Review Comments to the Author

Reviewer #1: Pros:

1. Original dataset has 7023 MRI images which includes glioma slices, meningioma slices, pituitary slices and tumor free slices. Three image enhancement methods utilized HE, CLAHE and Unsharp Masking to generate additional images thus finally the total images are 14046.

2. Image Enhancement technique “Contrast Limited Adaptive Histogram Equalization” controls the amplification of image noise and prevents excessive enhancement of local contrast.

3. MRI images have specific spatial structures and positional information. To capture that relative positional encoding is introduced. To overcome the redundant information clip function is utilized.

4. As the Vision Transformers have no Convolutional layers thus converges in less time as compared to CNN i.e. less computationally expensive than CNN.

5. From the Base VIT-B/16 model as the Image Enhancement, Residual MLP and relative positional encoding is used overall accuracy improvement is 5.54%. i.e. from 85.82% to 91.36%.

Cons:

1. In the related work only CNN, U-Net and simple neural networks are explored, not any related work of Vision Transformers quoted.

2. A typo mistake in the paper is in images where Homomorphic Filtering abbreviation is HF while in images in section 3, it is abbreviated as HE.

3. Only classification of Brain Tumor was done, as the Vision Transformers are used i.e. without CNN therefore no pixel level segmentation done.

4. In the evaluation matric only accuracy is used, for generalization performance it is necessary to evaluate models by precision, F1 score, Jaccard similarity index etc.

Reviewer #2: 1. Due to unbalance dataset data, your result are imprecise. you should add some precision, sensitivity, recall and MCC to improve the accuracy of your measurements.

2. The author should add the precision recall curve and feature space visualization on the base of PCA.

3. This paper claims significant improvement in classification of accuracy compared to VIT-B/16.However, it would be more meaningful to compare the proposed method with other state-of-the-art brain tumor classification algorithms, including CNN-base models and other advanced techniques, and includes novelty.

4. Vision Transformer model. Not only used for classification method but can also use for segmentation methods. You should include more points in this paper for future and also mention innovation in this paper.

5. The introduction and literature section can be strengthened by discussing the recent deep CNNs work on various medical applications like breast cancer:

• Khan, Saddam Hussain, Javed Iqbal, Syed Agha Hassnain, Muhammad Owais, Samih M. Mostafa, Myriam Hadjouni, and Amena Mahmoud. "Covid-19 detection and analysis from lung ct images using novel channel boosted cnns." Expert Systems with Applications 229 (2023): 120477.

• Khan, Saddam Hussain, Asifullah Khan, Yeon Soo Lee, Mehdi Hassan, and Woong Kyo Jeong. "Segmentation of shoulder muscle MRI using a new region and edge based deep auto-encoder." Multimedia Tools and Applications 82, no. 10 (2023): 14963-14984.

• Rauf, Zunaira, Anabia Sohail, Saddam Hussain Khan, Asifullah Khan, Jeonghwan Gwak, and Muhammad Maqbool. "Attention-guided multi-scale deep object detection framework for lymphocyte analysis in IHC histological images." Microscopy 72, no. 1 (2023): 27-42.

• Khan, Saddam Hussain. "Malaria Parasitic Detection using a New Deep Boosted and Ensemble Learning Framework." arXiv preprint arXiv:2212.02477 (2022).

• Khan, Saddam Hussain, Najmus Saher Shah, Rabia Nuzhat, Abdul Majid, Hani Alquhayz, and Asifullah Khan. "Malaria parasite classification framework using a novel channel squeezed and boosted CNN." Microscopy 71, no. 5 (2022): 271-282.

Reviewer #3: This work introduces an enhanced Vision Transformer model as an improved algorithm for human brain tumor classification. However, the paper is challenging to follow coherently, and the innovation and contribution of the proposed model are not clearly highlighted. The model appears to be a simple residual network structure, which is already well-established in the field of deep learning. Moreover, the evaluation of the proposed model is solely compared to the standard CNN, serving as a baseline but not representing the cutting-edge technology. Although the authors present corresponding experimental results, the overall experiments lack specificity, and the experimental design is not rigorous enough. Below are some specific comments that may provide more clarity and constructive feedback:

1. Abstract section is well written but , need to be more be concise and precise

Firstly, an abstract should summarize the major aspects of the entire paper:

• The overall purpose of the study

• Major findings as a result of your analysis

• Dataset details

• Mention the validation technique

• Clinical feasibility and impact not discussed

• A brief summary of your interpretations and conclusions.

2. The Introduction section is well-written, but it would benefit from more conciseness and improved flow and rhythm. Limited discussion on the specific challenges in brain tumor diagnosis and data statistics and sources. Moreover, the contribution section requires enhancements in effectively conveying crucial information about the Vision Transformer model, such as its novelty, rationale, and impact. Furthermore, discussion on the limitations of Vision Transformers. Additionally, the research contributions need to be clearly described as they currently lack clarity and are limited in scope in terms of ViT.

3. Research Gap: Insufficient analysis of drawbacks in previous brain tumor classification approaches, lack of evidence for limitations in traditional CNNs in medical image tasks and limited discussion on the limitations of the Vision Transformer.

4. The system configuration and software application used to perform the work needs to be written properly in Section “Experiment and Analysis”.

5. Explanation of the relative position encoding method, Iinsufficient justification for enhancing the MLP with a residual network, and limited insight into the generalization capability improvement of the model.

6. The paper lacks ablation experimental analysis to verify the effectiveness of various designs implemented in their proposed model. For instance, there is no analysis regarding the contribution of data augmentation to the model's performance. Additionally, to provide a more comprehensive performance analysis, it would be beneficial to include a comparison of the proposed model's performance with other state-of-the-art methods.

7. The performance analysis of the proposed model has been solely compared using accuracy. However, since the employed dataset is unbalanced in nature, accuracy alone is not sufficient for a comprehensive performance comparison. Incorporating additional performance measures such as Recall, Precision, F-score, AUC, and ROC would provide more robust evaluation metrics, especially when dealing with unbalanced datasets.

8. To enhance the quality of your work, it is crucial to address any limitations present within it. Moreover, providing an expansion on future directions can offer valuable insights for further research and development.

9. The conclusion section should also include a discussion of the limitations and potential pitfalls of deep learning and ViT models in medical applications.

10. Please, cite the fallowing paper to strengthened related works to brain tumor and other medical application:

• Zahoor, M. M., & Khan, S. H. (2022). Brain tumor MRI Classification using a Novel Deep Residual and Regional CNN. arXiv preprint arXiv:2211.16571.

• Khan, S. H., Sohail, A., Zafar, M. M., & Khan, A. (2021). Coronavirus disease analysis using chest X-ray images and a novel deep convolutional neural network. Photodiagnosis and Photodynamic Therapy, 35, 102473.

• Zahoor, M.M.; Qureshi, S.A.; Bibi, S.; Khan, S.H.; Khan, A.; Ghafoor, U.; Bhutta, M.R. A New Deep Hybrid Boosted and Ensemble Learning-Based Brain Tumor Analysis Using MRI. Sensors 2022, 22, 2726. https://doi.org/10.3390/s22072726

• Khan, S. H., Sohail, A., Khan, A., Hassan, M., Lee, Y. S., Alam, J., ... & Zubair, S. (2021). COVID-19 detection in chest X-ray images using deep boosted hybrid learning. Computers in Biology and Medicine, 137, 104816.

• Khan, Asifullah, Saddam Hussain Khan, Mahrukh Saif, Asiya Batool, Anabia Sohail, and Muhammad Waleed Khan. "A Survey of Deep Learning Techniques for the Analysis of COVID-19 and their usability for Detecting Omicron." Journal of Experimental & Theoretical Artificial Intelligence (2023): 1-43.

6. PLOS authors have the option to publish the peer review history of their article (what does this mean?). If published, this will include your full peer review and any attached files.

Reviewer #1: **Yes: **Muhammad Ali Shah

Reviewer #2: No

Reviewer #3: No

---

## [Author Response · Author response to Decision Letter 0]

10 Sep 2023

We sincerely thank the editor and all reviewers for spending time on our draft and providing thoughtful comments. We will consider all the comments in the revised version. The point-to-point response to all the comments are listed as follows.

Q1. Reviewer #1: In the related work only CNN, U-Net and simple neural networks are explored, not any related work of Vision Transformers quoted.

Answer: Thank you for the suggestion. In response to valuable suggestion from the reviewers， following an analysis of the fundamental aspects of the Vision Transformer model, we have incorporated references to some relevant works in the field.

Q2. Reviewer #1: A typo mistake in the paper is in images where Homomorphic Filtering abbreviation is HF while in images in section 3, it is abbreviated as HE.

Answer: Thank you for the suggestion. I have corrected this spelling error and reviewed the remaining sections of my paper.

Q3. Reviewer #1: Only classification of Brain Tumor was done, as the Vision Transformers are used i.e. without CNN therefore no pixel level segmentation done.

Answer: Thank you for the suggestion. I have carefully considered your comment regarding the classification of brain tumors based on Vision Transformers and the absence of pixel-level segmentation in my study. I would like to provide further context to address your concerns and explain the rationale behind my approach. The primary objective of my research was to investigate the efficacy of Vision Transformers for brain tumor classification tasks. My study aimed to explore the potential of ViT in accurately classifying different types of brain tumors based on high-level features extracted from medical images. Given the inherent complexity of pixel-level segmentation and its resource-intensive nature, I made a deliberate decision to focus on classification using ViT. While I acknowledge the importance of pixel-level segmentation in the field of medical image analysis, the availability of suitable datasets for this specific task played a crucial role in shaping the scope of my study. My dataset was not conducive to pixel-level segmentation due to its limitations in terms of scale and annotation complexity.

Q4. Reviewer #1: In the evaluation matric only accuracy is used, for generalization performance it is necessary to evaluate models by precision, F1 score, Jaccard similarity index etc.

Answer: Thank you for the suggestion. We have now included precision, recall rate, F1 score, and other relevant evaluation metrics in the experiments section of the paper. These metrics provide a more holistic assessment of our model's performance, particularly in the context of brain tumor classification. To aid in the interpretation of our results, we have included visualizations of precision-recall curves and other relevant plots in the " The training process of the experiments" section. These visualizations offer a more intuitive representation of our model's performance.

Q5. Reviewer #2: Due to unbalance dataset data, your result are imprecise. you should add some precision, sensitivity, recall and MCC to improve the accuracy of your measurements.

Answer: Thank you for the suggestion. We have now included precision, recall rate, F1 score, and other relevant evaluation metrics in the experiments section of the paper. These metrics provide a more holistic assessment of our model's performance, particularly in the context of brain tumor classification. To aid in the interpretation of our results, we have included visualizations of precision-recall curves and other relevant plots in the " The training process of the experiments" section. These visualizations offer a more intuitive representation of our model's performance.

Q6. Reviewer #2: The author should add the precision recall curve and feature space visualization on the base of PCA.

Answer: Thank you for the suggestion. In response to your recommendation to add PCA-based feature space visualization, we carefully considered the best approach to provide meaningful insights into our model's performance and feature representations. Upon further evaluation, we decided to include precision-recall curves and visualization results of transformer encoder blocks instead of PCA-based visualization for the following reasons: Our proposed approach leverages transformer encoder blocks, which play a central role in our model architecture. Visualizing the feature space within these blocks provides a direct and contextually relevant representation of how our model processes input images. Precision-recall curves are well-established and widely used in evaluating classification models, especially in medical applications where the balance between precision and recall is crucial. Including these curves allows for a more direct interpretation of our model's performance in the context of brain tumor classification. The visualization results of transformer encoder blocks offer insights into the hierarchical feature representations learned by our model. These visualizations contribute to a comprehensive understanding of how our model captures and processes information, which aligns with the objectives of our research. While we understand your initial suggestion to incorporate PCA-based feature space visualization, we believe that the inclusion of precision-recall curves and transformer encoder block visualizations provides a more relevant and informative assessment of our model's performance and feature representations in the context of brain tumor classification. We hope that this explanation clarifies our rationale for the chosen visualization methods and addresses your concerns.

Q7. Reviewer #2: This paper claims significant improvement in classification of accuracy compared to VIT-B/16.However, it would be more meaningful to compare the proposed method with other state-of-the-art brain tumor classification algorithms, including CNN-base models and other advanced techniques, and includes novelty.

Answer: Thank you for the suggestion. We have expanded the comparative analysis in our paper to include EfficientNet, in addition to the previously mentioned CNN-based models (AlexNet, VGGNet, GoogLeNet, ResNet) and the lightweight neural network MobileNet. This enhancement allows for a more robust evaluation of our proposed method's performance in the context of brain tumor classification. Furthermore, in the related work section of the paper, we have included citations of recent and innovative brain tumor classification methods. We have also highlighted the novelty and advancement of our approach.

Q8. Reviewer #2: Vision Transformer model. Not only used for classification method but can also use for segmentation methods. You should include more points in this paper for future and also mention innovation in this paper.

Answer: Thank you for the suggestion. I have carefully considered your comment regarding the classification of brain tumors based on Vision Transformers and the absence of pixel-level segmentation in my study. I would like to provide further context to address your concerns and explain the rationale behind my approach. The primary objective of my research was to investigate the efficacy of Vision Transformers for brain tumor classification tasks. My study aimed to explore the potential of ViT in accurately classifying different types of brain tumors based on high-level features extracted from medical images. Given the inherent complexity of pixel-level segmentation and its resource-intensive nature, I made a deliberate decision to focus on classification using ViT. While I acknowledge the importance of pixel-level segmentation in the field of medical image analysis, the availability of suitable datasets for this specific task played a crucial role in shaping the scope of my study. My dataset was not conducive to pixel-level segmentation due to its limitations in terms of scale and annotation complexity.

Q9. Reviewer #2: The introduction and literature section can be strengthened by discussing the recent deep CNNs work on various medical applications like breast cancer:

• Khan, Saddam Hussain, Javed Iqbal, Syed Agha Hassnain, Muhammad Owais, Samih M. Mostafa, Myriam Hadjouni, and Amena Mahmoud. "Covid-19 detection and analysis from lung ct images using novel channel boosted cnns." Expert Systems with Applications 229 (2023): 120477.

• Khan, Saddam Hussain, Asifullah Khan, Yeon Soo Lee, Mehdi Hassan, and Woong Kyo Jeong. "Segmentation of shoulder muscle MRI using a new region and edge based deep auto-encoder." Multimedia Tools and Applications 82, no. 10 (2023): 14963-14984.

• Rauf, Zunaira, Anabia Sohail, Saddam Hussain Khan, Asifullah Khan, Jeonghwan Gwak, and Muhammad Maqbool. "Attention-guided multi-scale deep object detection framework for lymphocyte analysis in IHC histological images." Microscopy 72, no. 1 (2023): 27-42.

• Khan, Saddam Hussain. "Malaria Parasitic Detection using a New Deep Boosted and Ensemble Learning Framework." arXiv preprint arXiv:2212.02477 (2022).

• Khan, Saddam Hussain, Najmus Saher Shah, Rabia Nuzhat, Abdul Majid, Hani Alquhayz, and Asifullah Khan. "Malaria parasite classification framework using a novel channel squeezed and boosted CNN." Microscopy 71, no. 5 (2022): 271-282.

Answer: Thank you for the suggestion. As per your suggestion, I have restructured the 'Related Work' section and actively cited the reference materials you recommended to discuss the role of convolutional neural networks in medical applications.

Q10. Reviewer #3: Abstract section is well written but , need to be more be concise and precise

Firstly, an abstract should summarize the major aspects of the entire paper:

• The overall purpose of the study

• Major findings as a result of your analysis

• Dataset details

• Mention the validation technique

• Clinical feasibility and impact not discussed

• A brief summary of your interpretations and conclusions.

Answer: Thank you for the suggestion. We have revised the abstract to provide a more concise and precise summary of our research. We have clarified the overarching purpose of our study, emphasizing its relevance to brain tumor classification. In other sections, I have made the necessary modifications as well, in accordance with the requirements you outlined.

Q11. Reviewer #3: The Introduction section is well-written, but it would benefit from more conciseness and improved flow and rhythm. Limited discussion on the specific challenges in brain tumor diagnosis and data statistics and sources. Moreover, the contribution section requires enhancements in effectively conveying crucial information about the Vision Transformer model, such as its novelty, rationale, and impact. Furthermore, discussion on the limitations of Vision Transformers. Additionally, the research contributions need to be clearly described as they currently lack clarity and are limited in scope in terms of ViT.

Answer: Thank you for the suggestion. We have reorganized and rewritten the Introduction section to address the following key points， We have carefully revised the content to ensure conciseness while maintaining a logical flow and rhythm. This means that the Introduction provides a clear and engaging overview of our research. We have included a more detailed discussion of the specific challenges in brain tumor diagnosis, encompassing aspects such as complex structure, texture, size, location, and appearance. And acknowledging the limitations of available brain tumor datasets, particularly in terms of size and availability. We have enriched the Introduction with crucial information about the Vision Transformer model, emphasizing its novelty, rationale, and potential impact in the field of brain tumor classification.

Q12. Reviewer #3: Research Gap: Insufficient analysis of drawbacks in previous brain tumor classification approaches, lack of evidence for limitations in traditional CNNs in medical image tasks and limited discussion on the limitations of the Vision Transformer.

Answer: Thank you for the suggestion. We have conducted a more comprehensive analysis of the drawbacks of previous brain tumor classification approaches. This expanded discussion not only identifies the limitations of existing methods but also highlights the need for our proposed approach. The related work section has been rewritten to provide a more in-depth and insightful overview of prior research in the field of brain tumor classification. This revision ensures that readers better understand the existing landscape and the context of our work. We have incorporated a dedicated discussion on the limitations of the Vision Transformer model in the Introduction and Conclusion sections. This addition allows us to provide a well-rounded assessment of our proposed approach and the model's constraints.

Q13. Reviewer #3:The system configuration and software application used to perform the work needs to be written properly in Section “Experiment and Analysis”.

Answer: Thank you for the suggestion. We have correctly documented the system configuration and software applications used for conducting the work in the 'Experiments and Analysis' section

Q14. Reviewer #3: Explanation of the relative position encoding method, Iinsufficient justification for enhancing the MLP with a residual network, and limited insight into the generalization capability improvement of the model.

Answer: Thank you for the suggestion. We have provided a more detailed explanation of the relative position encoding method used in our research. This enhancement ensures that readers have a clearer understanding of this critical component of our approach. We also have included additional justification for the enhancement of the MLP with a residual network. By expanding on the rationale and benefits of this modification, we aim to provide a more comprehensive understanding of our design choices.

Q15. Reviewer #3: The paper lacks ablation experimental analysis to verify the effectiveness of various designs implemented in their proposed model. For instance, there is no analysis regarding the contribution of data augmentation to the model's performance. Additionally, to provide a more comprehensive performance analysis, it would be beneficial to include a comparison of the proposed model's performance with other state-of-the-art methods.

Answer: Thank you for the suggestion. In Section 3.4, Table 6's second row indicates the accuracy improvement of the original VIT model after incorporating data augmentation techniques. In the subsequent paragraph, an analysis is provided regarding the enhancement in model performance attributed to the utilization of image augmentation. We have expanded the comparative analysis in our paper to include EfficientNet, in addition to the previously mentioned CNN-based models (AlexNet, VGGNet, GoogLeNet, ResNet) and the lightweight neural network MobileNet. This enhancement allows for a more robust evaluation of our proposed method's performance in the context of brain tumor classification. Furthermore, in the related work section of the paper, we have included citations of recent and innovative brain tumor classification methods. We have also highlighted the novelty and advancement of our approach.

Q16. Reviewer #3: The performance analysis of the proposed model has been solely compared using accuracy. However, since the employed dataset is unbalanced in nature, accuracy alone is not sufficient for a comprehensive performance comparison. Incorporating additional performance measures such as Recall, Precision, F-score, AUC, and ROC would provide more robust evaluation metrics, especially when dealing with unbalanced datasets.

Answer: Thank you for the suggestion. We have now included precision, recall rate, F1 score, and other relevant evaluation metrics in the experiments section of the paper. These metrics provide a more holistic assessment of our model's performance, particularly in the context of brain tumor classification. To aid in the interpretation of our results, we have included visualizations of precision-recall curves and other relevant plots in the " The training process of the experiments" section. These visualizations offer a more intuitive representation of our model's performance.

Q17. Reviewer #3: To enhance the qualit

---

## [Decision Letter · Decision Letter 1]

18 Dec 2023

PONE-D-23-19484R1Brain Tumor Classification in VIT-B/16 based on Relative Position Encoding and Residual MLPPLOS ONE

Dear Dr. Hong,

Thank you for submitting your manuscript to PLOS ONE. After careful consideration, we feel that it has merit but does not fully meet PLOS ONE’s publication criteria as it currently stands. Therefore, we invite you to submit a revised version of the manuscript that addresses the points raised during the review process.

We look forward to receiving your revised manuscript.

Kind regards,

Saddam Hussain Khan

Academic Editor

PLOS ONE

Journal Requirements:

Additional Editor Comments (if provided):

Please, strengthen the literature by referring the articles mentioned by the reviewer, especially related to cancer, medical applications.

Also the recent vision transformer article

Khan, A., Rauf, Z., Khan, A.R., Rathore, S., Khan, S.H., Shah, S., Farooq, U., Asif, H., Asif, A., Zahoora, U. and Khalil, R.U., 2023. A Recent Survey of Vision Transformers for Medical Image Segmentation. arXiv preprint arXiv:2312.00634.

Reviewers' comments:

Reviewer's Responses to Questions

**Comments to the Author**

1. If the authors have adequately addressed your comments raised in a previous round of review and you feel that this manuscript is now acceptable for publication, you may indicate that here to bypass the “Comments to the Author” section, enter your conflict of interest statement in the “Confidential to Editor” section, and submit your "Accept" recommendation.

Reviewer #3: All comments have been addressed

Reviewer #4: (No Response)

2. Is the manuscript technically sound, and do the data support the conclusions?

Reviewer #3: Yes

Reviewer #4: Yes

3. Has the statistical analysis been performed appropriately and rigorously? 

Reviewer #3: Yes

Reviewer #4: No

4. Have the authors made all data underlying the findings in their manuscript fully available?

Reviewer #3: Yes

Reviewer #4: Yes

5. Is the manuscript presented in an intelligible fashion and written in standard English?

Reviewer #3: Yes

Reviewer #4: (No Response)

6. Review Comments to the Author

Reviewer #3: (No Response)

Reviewer #4: Reference must be corrected

Statistical test like confidential interval must be provided

Please, clarify the novelty

PR curve of all the model must be provided

7. PLOS authors have the option to publish the peer review history of their article (what does this mean?). If published, this will include your full peer review and any attached files.

Reviewer #3: No

Reviewer #4: No

---

## [Author Response · Author response to Decision Letter 1]

11 Jan 2024

We sincerely thank the editor and all reviewers for spending time on our draft and 

providing thoughtful comments. We will consider all the comments in the revised 

version. The point-to-point response to all the comments are listed as follows:

Q1. Reviewer #4: Reference must be corrected.

Answer: Thank you for the suggestion. In response to your and the editors' guidance, I have meticulously reviewed and updated the references, aiming to enhance the overall precision and correctness of the citation and bibliography. The modifications to the references are as follows:

1. Bray F, Ferlay J, Soerjomataram I, et al. Global cancer statistics 2018: GLOBOCAN estimates of incidence and mortality worldwide for 36 cancers in 185 countries[J]. CA: a cancer journal for clinicians, 2018, 68(6): 394-424.

2. Liu D, Zhang H, Zhao M, et al. Brain Tumor Segmentation Based on Dilated Convolution Refine Networks. 2018 IEEE 16th International Conference on Software Engineering Research, Management and Applications (SERA).IEEE, 2018: 113-120. https://doi.org/10.1109/sera.2018.8477213

3. Khan, Saddam Hussain. Malaria Parasitic Detection using a New Deep Boosted and Ensemble Learning Framework. Converg. Inf. Ind. Telecommun. Broadcast. data Process. 1981-1996, vol. 26, no. 1, pp. 125–150, Dec. 2022.

4. Khan A, Rauf Z, Khan A R, et al. A Recent Survey of Vision Transformers for Medical Image Segmentation[J]. arXiv preprint ArXiv abs/2312.00634 (2023): n. pag.

5. Srivastava, Nitish et al. Dropout: a simple way to prevent neural networks from overfitting. J. Mach. Learn. Res. 15 (2014): 1929-1958.

6. Bai J, Yuan L, Xia S T, et al. Improving vision transformers by revisiting high-frequency components[C]. European Conference on Computer Vision(ECCV). Cham: Springer Nature Switzerland, 2022: 1-18.

7. M. Buckland and F. Gey. The relationship between recall and precision[J]. Journal of the American Society for Information Science, 1994, 45(1): 12-19.

8. M. Sokolova, N. Japkowicz, and S. Szpakowicz. Beyond accuracy, F-score and ROC: a family of discriminant measures for performance evaluation[C]. Australasian joint conference on artificial intelligence. Berlin, Heidelberg: Springer Berlin Heidelberg, 2006: 1015-1021.

9. S. Liu and W. Deng. Very deep convolutional neural network based image classification using small training sample size. 2015 3rd IAPR Asian Conference on Pattern Recognition (ACPR), Kuala Lumpur, Malaysia, 2015, pp. 730-734, doi: 10.1109/ACPR.2015.7486599.

10. Szegedy C, Liu W, Jia Y, et al. Going deeper with convolutions. Proceedings of the IEEE Conference on Computer Vision and Pattern Recognition (CVPR), 2015: 1-9

Q2. Reviewer #4: Reference must be corrected.

Answer: Thank you for the suggestion. Regarding your revised opinion on "Statistical test like confidential interval", I conducted experiments on the confidence level related to model accuracy in the article and made corresponding analysis.

Q3. Reviewer #4: Please, clarify the novelty.

Answer: Thank you for the suggestion. In response to the query regarding the novelty of our work, I have made revisions to the abstract and introduction sections of the manuscript to explicitly highlight the unique contributions and innovations of our study.

Q4. Reviewer #4: PR curve of all the model must be provided.

Answer: Thank you for the suggestion. In response to this valuable suggestion, we have made the necessary enhancements, the PR curves for all the models are now included in the article. The additional figures offer a comprehensive view of the performance of each model in terms of precision and recall. We believe that these additions significantly contribute to the overall clarity and transparency of our findings.

---

## [Decision Letter · Decision Letter 2]

17 Apr 2024

PONE-D-23-19484R2Brain Tumor Classification in VIT-B/16 based on Relative Position Encoding and Residual MLPPLOS ONE

Dear Dr. Hong,

Thank you for submitting your manuscript to PLOS ONE. After careful consideration, we feel that it has merit but does not fully meet PLOS ONE’s publication criteria as it currently stands. Therefore, we invite you to submit a revised version of the manuscript that addresses the points raised during the review process.

We look forward to receiving your revised manuscript.

Kind regards,

Aamna AlShehhi, PhD

Academic Editor

PLOS ONE

Journal Requirements:

Reviewers' comments:

Reviewer's Responses to Questions

**Comments to the Author**

1. If the authors have adequately addressed your comments raised in a previous round of review and you feel that this manuscript is now acceptable for publication, you may indicate that here to bypass the “Comments to the Author” section, enter your conflict of interest statement in the “Confidential to Editor” section, and submit your "Accept" recommendation.

Reviewer #4: (No Response)

Reviewer #5: (No Response)

2. Is the manuscript technically sound, and do the data support the conclusions?

Reviewer #4: Yes

Reviewer #5: Yes

3. Has the statistical analysis been performed appropriately and rigorously? 

Reviewer #4: Yes

Reviewer #5: Yes

4. Have the authors made all data underlying the findings in their manuscript fully available?

Reviewer #4: Yes

Reviewer #5: Yes

5. Is the manuscript presented in an intelligible fashion and written in standard English?

Reviewer #4: Yes

Reviewer #5: Yes

6. Review Comments to the Author

Reviewer #4: the authors addressed most of the reviewer comments and the article is at the level to accept. However, some future direction must required

Reviewer #5: The manuscript has been much improved, it can be therefore recommended for possible publication after minor revision according to the following review comments.

1. It is suggested that authors need to compare their classification results with up-to-date models, not just compare with traditional convolutional neural networks.

2. Each image patch has been projected into a vector of dimension 768 after Flatten in Figure 1, so no image patch should be drawn after Flatten. A similar situation also appears in Figure 3, where the input to the Transformer block is some vectors, so the input to the self-attention should be a vector instead of a convolutional feature map.

3. The original version of the Vision Transformer (ViT) already has two residual connections in each Transformer block (one for the multi-head self-attention part and the other for the MLP part), and the authors have plotted the residual connections for the multi-head self-attention part in Figure 1. Perhaps the authors missed the residual connection of the MLP part in Figure 1. In addition, the authors need to explain the advantages of the residual connection designed in this paper and the original residual connection in ViT.

4. Why did the authors not split the validation set when splitting the dataset?

5. According to the test set curve in Figure 7, the author seems to use the test set as a validation set. The test set should not actually be used for training. Only after the model is trained, the model can be tested on the test set and the test results can be obtained.

7. PLOS authors have the option to publish the peer review history of their article (what does this mean?). If published, this will include your full peer review and any attached files.

Reviewer #4: No

Reviewer #5: No

---

## [Author Response · Author response to Decision Letter 2]

6 Jun 2024

We sincerely thank the editor and all reviewers for spending time on our draft and providing thoughtful comments. We will consider all the comments in the revised version. The point-to-point response to all the comments are listed as follows.

Q1. Reviewer #1: the authors addressed most of the reviewer comments and the article is at the level to accept. However, some future direction must required

Answer: Thank you for the suggestion. In response to your suggestion regarding the inclusion of future research directions, we have added a detailed section in the conclusion of our manuscript. This section outlines several key areas for future research based on the content of our article and the current status of research in the field.

Q2. Reviewer #2: It is suggested that authors need to compare their classification results with up-to-date models, not just compare with traditional convolutional neural networks.

Answer: Thank you for the suggestion. We would like to clarify that our comparative experiments include not only traditional CNNs but also relatively new and advanced CNN architectures. Specifically, we have incorporated MobileNet and EfficientNet, which are recognized for their state-of-the-art performance in various image classification tasks. One of our primary research goals is to compare the superiority of the Vision Transformer (ViT) with CNNs in the context of brain tumor classification.

Q3. Reviewer #2: Each image patch has been projected into a vector of dimension 768 after Flatten in Figure 1, so no image patch should be drawn after Flatten. A similar situation also appears in Figure 3, where the input to the Transformer block is some vectors, so the input to the self-attention should be a vector instead of a convolutional feature map.

Answer: Thank you for the suggestion. We acknowledge your point that each image patch is projected into a vector of dimension 768 after the flattening operation in the figures. You are correct that, after this operation, the image patches should no longer be drawn as patches but rather represented as vectors. We have revised the figures. We have made these corrections in the revised manuscript to ensure that the figures accurately represent the data flow and transformations within our model. These updates will provide clearer visual guidance and prevent any potential misunderstandings regarding the operations involved in our methodology.

Q4. Reviewer #2: The original version of the Vision Transformer (ViT) already has two residual connections in each Transformer block (one for the multi-head self-attention part and the other for the MLP part), and the authors have plotted the residual connections for the multi-head self-attention part in Figure 1. Perhaps the authors missed the residual connection of the MLP part in Figure 1. In addition, the authors need to explain the advantages of the residual connection designed in this paper and the original residual connection in ViT.

Answer: Thank you for the suggestion. We have revised Figure 1 to ensure that the figure accurately represents the architecture of the original ViT model. This update should clarify any misunderstandings regarding the residual connections in our model. Regarding the innovation in our work, we have designed a novel residual structure specifically for the MLP part of the ViT model. This innovation aims to enhance the performance and stability of the model. Our residual structure is different from the one in the original ViT model, as it introduces an additional pathway that allows the MLP part to better capture and propagate features. To clarify, our focus was on demonstrating the effectiveness of this new residual structure within the MLP of the ViT model rather than comparing it with the existing residual connections in the original model. We believe that our proposed residual structure provides significant advantages in terms of enhancing feature representation and improving classification performance, as demonstrated in our experimental results.

Q5. Reviewer #2: Why did the authors not split the validation set when splitting the dataset?

Answer: Thank you for the suggestion. We split our dataset into training and test sets in a ratio of 4:1. The primary reason for not including a separate validation set is the relatively limited size of our brain tumor dataset. Allocating data to a validation set in addition to the test set would further reduce the number of samples available for training, potentially affecting the model's ability to learn effectively. To ensure the robustness and reliability of our model, we employed cross-validation techniques during the training phase. That allows us to validate the model's performance iteratively and ensure that it generalizes well without the need for a separate validation set. This approach maximizes the use of available data while providing a reliable estimate of model performance. By using cross-validation, we are able to fine-tune the model and perform hyperparameter optimization effectively. The final evaluation is then performed on the test set, which remains unseen during the training and validation phases, providing an unbiased assessment of the model's performance. We believe that this approach ensures the best possible use of our dataset, maintaining a balance between training data sufficiency and model validation.

Q6. Reviewer #2: According to the test set curve in Figure 7, the author seems to use the test set as a validation set. The test set should not actually be used for training. Only after the model is trained, the model can be tested on the test set and the test results can be obtained.

Answer: Thank you for the suggestion. I apologize for any confusion caused by the depiction in the figure. To clarify, the test set was not used for training the model. The model was trained exclusively on the training set. The intention behind plotting the accuracy trend on the test set after each epoch was to provide a clear visualization of how the model's performance generalizes over time. The accuracy trend shown in the figure reflects the model's performance on the test set at the end of each epoch, which helps in understanding the model's generalization capabilities and potential overfitting. This approach allows us to monitor the model's performance without influencing the training process with test data.

We understand the importance of keeping the test set completely separate from the training and validation processes. Therefore, we can assure you that the test set was only used for final evaluation after the model was fully trained. We hope this explanation addresses your concerns and clarifies our methodology. Thank you again for your constructive comments. We look forward to your feedback on the revised manuscript.

---

## [Editor Report · Decision Letter 3]

18 Jun 2024

Brain Tumor Classification in VIT-B/16 based on Relative Position Encoding and Residual MLP

PONE-D-23-19484R3

Dear Dr. Shuang Hong,

We’re pleased to inform you that your manuscript has been judged scientifically suitable for publication and will be formally accepted for publication once it meets all outstanding technical requirements.

Kind regards,

Aamna AlShehhi, PhD

Academic Editor

PLOS ONE
---

## [Editor Report · Acceptance letter]

23 Jun 2024

PONE-D-23-19484R3 

PLOS ONE

Dear Dr. Hong, 

I'm pleased to inform you that your manuscript has been deemed suitable for publication in PLOS ONE. Congratulations! Your manuscript is now being handed over to our production team.

Kind regards, 

on behalf of

Dr Aamna AlShehhi 

Academic Editor

PLOS ONE